# Debias Coarsely, Sample Conditionally: Statistical Downscaling through Optimal Transport and Probabilistic Diffusion Models

**Zhong Yi Wan**[*]
Google Research
Mountain View, CA 94043, USA
`wanzy@google.com`

**Ricardo Baptista**[*]
California Institute of Technology
Pasadena, CA 91106, USA
`rsb@caltech.edu`

**Yi-fan Chen**
Google Research
Mountain View, CA 94043, USA
`yifanchen@google.com`

**John Roberts Anderson**
Google Research
Mountain View, CA 94043, USA
`janders@google.com`

**Anudhyan Boral**
Google Research
Mountain View, CA 94043, USA
`anudhyan@google.com`

**Fei Sha**
Google Research
Mountain View, CA 94043, USA
`fsha@google.com`

**Leonardo Zepeda-Núñez**
Google Research
Mountain View, CA 94043, USA
`lzepedanunez@google.com`

## Abstract

We introduce a two-stage probabilistic framework for statistical downscaling *using unpaired data*. Statistical downscaling seeks a probabilistic map to transform low-resolution data from a *biased* coarse-grained numerical scheme to high-resolution data that is consistent with a high-fidelity scheme. Our framework tackles the problem by composing two transformations: (i) a debiasing step via an optimal transport map, and (ii) an upsampling step achieved by a probabilistic diffusion model with *a posteriori* conditional sampling. This approach characterizes a conditional distribution *without needing paired data*, and faithfully recovers relevant physical statistics from biased samples. We demonstrate the utility of the proposed approach on one- and two-dimensional fluid flow problems, which are representative of the core difficulties present in numerical simulations of weather and climate. Our method produces realistic high-resolution outputs from low-resolution inputs, by upsampling resolutions of $8\times$ and $16\times$. Moreover, our procedure correctly matches the statistics of physical quantities, even when the low-frequency content of the inputs and outputs do not match, a crucial but difficult-to-satisfy assumption needed by current state-of-the-art alternatives. Code for this work is available at: `https://github.com/google-research/swirl-dynamics/tree/main/swirl_dynamics/projects/probabilistic_diffusion`.

---

[*]Equal contribution

37th Conference on Neural Information Processing Systems (NeurIPS 2023).

# 1   Introduction

Statistical downscaling is crucial to understanding and correlating simulations of complex dynamical systems at multiple resolutions. For example, in climate modeling, the computational complexity of general circulation models (GCMs) [4] grows rapidly with resolution. This severely limits the resolution of long-running climate simulations. Consequently, accurate predictions (as in forecasting localized, regional and short-term weather conditions) need to be *downscaled* from coarser lower-resolution models' outputs. This is a challenging task: coarser models do not resolve small-scale dynamics, thus creating bias [16, 69, 84]. They also lack the necessary physical details (for instance, regional weather depends heavily on local topography) to be of practical use for regional or local climate impact studies [33, 36], such as the prediction or risk assessment of extreme flooding [35, 44], heat waves [59], or wildfires [1].

At the most abstract level, *statistical downscaling* [81, 82] learns a map from low- to high-resolution data. However, it has several unique challenges. First, unlike supervised machine learning (ML), there is *no natural pairing of samples* from the low-resolution model (such as climate models [23]) with samples from higher-resolution ones (such as weather models that assimilate observations [40]). Even in simplified cases of idealized fluids problems, one cannot naively align the simulations in time, due to the chaotic behavior of the models: two simulations with very close initial conditions will diverge rapidly. Several recent studies in climate sciences have relied on synthetically generated paired datasets. The synthesis process, however, requires accessing both low- and high-resolution models and either (re)running costly high-resolution models while respecting the physical quantities in the low-resolution simulations [25, 43] or (re)running low-resolution models with additional terms nudging the outputs towards high-resolution trajectories [12]. In short, requiring data in correspondence for training severely limits the potential applicability of supervised ML methodologies in practice, despite their promising results [37, 39, 60, 66, 38].

Second, unlike the setting of (image) super-resolution [26], in which an ML model learns the (pseudo) inverse of a downsampling operator [13, 78], downscaling additionally needs to correct the bias. This difference is depicted in Fig. 1(a). Super-resolution can be recast as frequency extrapolation [10], in which the model reconstructs high-frequency contents, while matching the low-frequency contents of a low-resolution input. However, the restriction of the target high-resolution data may not match the distribution of the low-resolution data in Fourier space [49]. Therefore, debiasing is necessary to correct the Fourier spectrum of the low-resolution input to render it admissible for the target distribution (moving solid red to solid blue lines with the dashed blue extrapolation in Fig. 1). Debiasing allows us to address the crucial yet challenging prerequisite of aligning the low-frequency statistics between the low- and high-resolution datasets.

Given these two difficulties, statistical downscaling should be more naturally framed as matching two probability distributions linked by an unknown map; such a map emerges from both distributions representing the same underlying physical system, albeit with different characterizations of the system's statistics at multiple spatial and temporal resolutions. The core challenge is then: *how do we structure the downscaling map so that the (probabilistic) matching can effectively remediate the bias introduced by the coarser, i.e., the low-resolution, data distribution?*

Thus, the main idea behind our work is to introduce a debiasing step so that the debiased (yet, still coarser) distribution is closer to the target distribution of the high-resolution data. This step results in an intermediate representation for the data that preserves the correct statistics needed in the follow-up step of upsampling to yield the high-resolution distribution. In contrast to recent works on distribution matching for unpaired image-to-image translation [86] and climate modeling [32], the additional structure our work imposes on learning the mapping prevents the bias in the low-resolution data from polluting the upsampling step. We review those approaches in §2 and compare to them in §4.

Concretely, we propose a new probabilistic formulation for the downscaling problem that handles *unpaired data* directly, based on a factorization of the unknown map linking both low- and high-resolution distributions. This factorization is depicted in Fig. 1(b). By appropriately restricting the maps in the factorization, we rewrite the downscaling map as the composition of two procedures: a debiasing step performed using an optimal transport map [21], which *couples the data distributions* and corrects the biases of the low-resolution snapshots; followed by an upsampling step performed using conditional probabilistic diffusion models, which have produced state-of-the-art results for image synthesis and flow construction [5, 52, 71, 73].

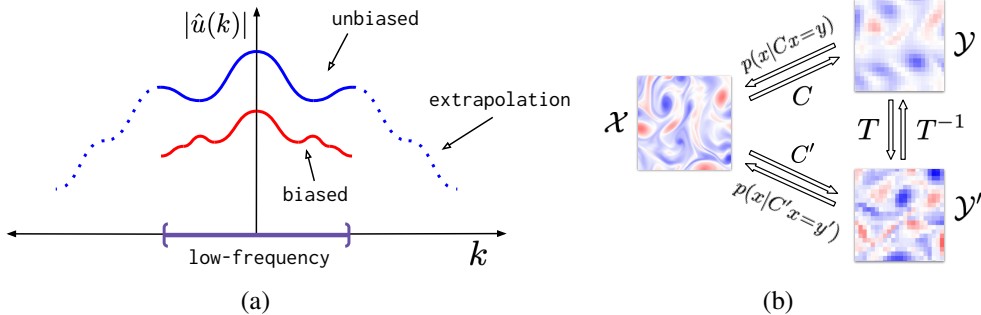

(a)                                   (b)

Figure 1: (a) Upsampling (super-resolution) as frequency extrapolation in the Fourier domain. The model extrapolates low-frequency content to higher-frequencies (dashed blue). The debiasing map corrects the biased low-frequency content (solid red). (b) Illustration of the proposed framework where $\mathcal{X}$ is the space of high-resolution data, $\mathcal{Y}$ is the space of low-resolution data, $C$ is an *unknown nonlinear* map linking $\mathcal{X}$ and $\mathcal{Y}$, $C'$ is a *known linear* downsampling map, $\mathcal{Y}'$ is an intermediate (low-resolution) space induced by the image of $C'$, and $T$ is an invertible debiasing map such that $C$ can be factorized as $T^{-1} \circ C'$. The conditional probabilities $p(x|C'x = y')$ are used for the probabilistic upsampling procedure.

We showcase the performance of our framework on idealized fluids problems that exhibit the same core difficulty present in atmospheric flows. We show that our framework is able to generate realistic snapshots that are faithful to the physical statistics, while outperforming several baselines.

## 2   Related work

The most direct approach to upsampling low-resolution data is to learn a low- to high-resolution mapping via paired data when it is possible to collect such data. For complex dynamical systems, several methods carefully manipulate high- and low-resolution models, either by nudging or by enforcing boundary conditions, to produce paired data without introducing spectral biases [12, 25]. Alternatively, if one has strong prior knowledge about the process of downsampling, optimization methods can solve an inverse problem to directly estimate the high-resolution data, leveraging prior assumptions such as sparsity in compressive sensing [9, 10] or translation invariance [42].

In our setting, there is no straightforward way to obtain paired data due to the nature of the problem (i.e., turbulent flows, with characteristically different statistics across a large span of spatio-temporal scales). In the weather and climate literature (see [79] for an extensive overview), prior knowledge can be exploited to downscale specific variables [81]. One of the most predominant methods of this type is bias-correction spatial disaggregation (BCSD), which combines traditional spline interpolation with a quantile matching bias correction [56], and linear models [41]. Recently, several studies have used ML to downscale physical quantities such as precipitation [78], but without quantifying the prediction uncertainty. Yet, a generally applicable method to downscale arbitrary variables is lacking.

Another difficulty is to remove the bias in the low resolution data. This is an instance of domain adaptation, a topic popularly studied in computer vision. Recent work has used generative models such as GANs and diffusion models to bridge the gap between two domains [5, 7, 14, 58, 60, 62, 68, 74, 83, 85]. A popular domain alignment method that was used in [32] for downscaling weather data is AlignFlow [34]. This approach learns normalizing flows for source and target data of the same dimension, and uses their common latent space to move across domains. The advantage of those methods is that they do not require training data from two domains in correspondence. Many of those approaches are related to optimal transport (OT), a rigorous mathematical framework for learning maps between two domains without paired data [80]. Recent computational advances in OT for discrete (i.e., empirical) measures [21, 64] have resulted in a wide set of methods for domain adaptation [20, 31]. Despite their empirical success with careful choices of regularization, their use alone for high-dimensional images has remained limited [61].

Our work uses diffusion models to perform upsampling after a debiasing step implemented with OT. We avoid common issues from GANs [75] and flow-based methods [54], which include over-smoothing, mode collapse and large model footprints [24, 52]. Also, due to the debiasing map, which matches the low-frequency content in distribution (see Fig. 1(a)), we do not need to explicitly impose that the low-frequency power spectra of the two datasets match like some competing methods do [5]. Compared to formulations that perform upsampling and debiasing simultaneously [5, 78], our

framework performs these two tasks separately, by only training (and independently validating) a single probabilistic diffusion model for the high-resolution data once. This allows us to quickly assess different modeling choices, such as the linear downsampling map, by combining the diffusion model with different debiasing maps. Lastly, in comparison to other two-stage approaches [5, 32], debiasing is conducted at low-resolutions, which is less expensive as it is performed on a much smaller space, and more efficient as it is not hampered from spurious biases introduced by interpolation techniques.

## 3  Methodology

**Setup**  We consider two spaces: the high-fidelity, high-resolution space $\mathcal{X} = \mathbb{R}^d$ and the low-fidelity, low-resolution space $\mathcal{Y} = \mathbb{R}^{d'}$, where we suppose that $d > d'$. We model the elements $X \in \mathcal{X}$ and $Y \in \mathcal{Y}$ as random variables with marginal distributions, $\mu_X$ and $\mu_Y$, respectively. In addition, we suppose there is a statistical model relating the $X$ and $Y$ variables via $C \colon \mathcal{X} \to \mathcal{Y}$, an unknown and possibly nonlinear, downsampling map. See Fig. 1(b) for a diagram.

Given an observed realization $\bar{y} \in \mathcal{Y}$, which we refer to as a *snapshot*, we formulate downscaling as the problem of sampling from the conditional probability distribution $p(x|E_{\bar{y}})$ for the event $E_{\bar{y}} := \{x \in \mathcal{X} \,|\, C(x) = \bar{y}\}$, which we denote by $p(x|C(x) = \bar{y})$. Our objective is to sample this distribution given only access to marginal samples of $X$ and $Y$.

**Main idea**  In general, downscaling is an ill-posed problem given that the joint distribution of $X$ and $Y$ is not prescribed by a known statistical model. Therefore, we seek an approximation to $C$ so the statistical properties of $X$ are preserved given samples of $\mu_Y$. In particular, such a map should satisfy $C_\sharp \mu_X = \mu_Y$, where $C_\sharp \mu_X$ denotes the push-forward measure of $\mu_X$ through $C$.

In this work, we impose a structured ansatz to approximate $C$. Specifically, we *factorize* the map $C$ as the composition of a known and linear *downsampling map* $C'$, and an invertible *debiasing map* $T$:

$$C = T^{-1} \circ C', \quad \text{such that} \quad (T^{-1} \circ C')_\sharp \mu_X = \mu_Y, \tag{1}$$

or alternatively, $C'_\sharp \mu_X = T_\sharp \mu_Y$. This factorization decouples and explicitly addresses two entangled goals in downscaling: debiasing and upsampling. We discuss the advantage of such factorization, after sketching how $C'$ and $T$ are implemented.

The range of the downsampling map $C' \colon \mathcal{X} \to \mathcal{Y}'$ defines an *intermediate* space $\mathcal{Y}' = \mathbb{R}^{d'}$ of high-fidelity low-resolution samples with measure $\mu_{Y'}$. Moreover, the joint space $\mathcal{X} \times \mathcal{Y}'$ is built by projecting samples of $X$ into $\mathcal{Y}'$, i.e., $(x, y') = (x, C'x) \in \mathcal{X} \times \mathcal{Y}'$; see Fig. 1(b). Using these spaces, we decompose the domain adaptation problem into the following three sub-problems:

1. *High-resolution prior*: Estimate the marginal density $p(x)$;
2. *Conditional modeling*: For the joint variables $X \times Y'$, approximate $p(x|C'x = y')$;
3. *Debiasing*: Compute a transport map such that $T_\sharp \mu_Y = C'_\sharp \mu_X$.

For the first sub-problem, we train an *unconditional* model to approximate $\mu_X$, or $p(x)$, as explained in §3.1. For the second sub-problem, we leverage the prior model and $y' \in \mathcal{Y}'$ to build a model for *a posteriori* conditional sampling of $p(x|C'x = y')$, which allows us to upsample snapshots from $\mathcal{Y}'$ to $\mathcal{X}$, as explained in §3.2. For the third sub-problem, we use domain adaptation to shift the resulting model from the source domain $\mathcal{X} \times \mathcal{Y}'$ to the target domain $\mathcal{X} \times \mathcal{Y}$, for which there is no labeled data. For such a task, we build a transport map $T : \mathcal{Y} \to \mathcal{Y}'$ satisfying the condition that $T_\sharp \mu_Y = \mu_{Y'} = C'_\sharp \mu_X$. This map is found by solving an optimal transport problem, which we explain in §3.3.

Lastly, we merge the solutions to the sub-problems to arrive at our core downscaling methodology, which is summarized in Alg. 1. In particular, given a low-fidelity and low-resolution sample $\bar{y}$, we use the optimal transport map $T$ to project the sample to the high-fidelity space $\bar{y}' = T(\bar{y})$ and use the conditional model to sample $p(x|C'x = \bar{y}')$. The resulting samples are contained in the high-fidelity and high-resolution space.

The factorization in Eq. (1) has several advantages. We do not require a cycle-consistency type of loss [34, 86]: the consistency condition is automatically enforced by Eq. (1) and the conditional sampling. By using a linear downsampling map $C'$, it is trivial to create the intermediate space $\mathcal{Y}'$, while rendering the conditional sampling tractable: conditional sampling with a nonlinear map is

---
**Algorithm 1** : **Downscaling from** $\bar{y} \in \mathcal{Y}$ **to** $\bar{x}(\bar{y}) \in \mathcal{X}$.
---
**Input: low-fidelity, low-resolution sample:** $y$
1. Compute the debiased term $\bar{y}' = T_\gamma(\bar{y})$ using the barycentric approximation in Eq. (11).
2. Modify the scoring function in Eq. (6) using the conditional denoiser in Eq. (7).
3. Solve the reverse SDE in Eq. (5), and obtain $\bar{x}$.
**Output: high-fidelity, high-resolution sample :** $\bar{x}$
---

often more expensive and it requires more involved tuning [17, 18]. The factorization also allows us to compute the debiasing map in a considerably lower dimensional space, which conveniently requires less data to cover the full distribution, and fewer iterations to find the optimal map [21].

### 3.1 High-resolution prior

To approximate the prior of the high-resolution snapshots we use a probabilistic diffusion model, which is known to avoid several drawbacks of other generative models used for super-resolution [52], while providing greater flexibility for *a posteriori* conditioning [17, 30, 46, 47].

Intuitively, diffusion-based generative models involves iteratively transforming samples from an initial noise distribution $p_T$ into ones from the target data distribution $p_0 = p_{\text{data}}$. Noise is removed sequentially such that samples follow a family of marginal distributions $p_t(x_t; \sigma_t)$ for decreasing diffusion times $t$ and noise levels $\sigma_t$. Conveniently, such distributions are given by a forward noising process that is described by the stochastic differential equation (SDE) [45, 73]

$$dx_t = f(x_t, t)dt + g(x_t, t)dW_t, \tag{2}$$

with drift $f$, diffusion coefficient $g$, and the standard Wiener process $W_t$. Following [45], we set

$$f(x_t, t) = f(t)x_t := \frac{\dot{s}_t}{s_t}x_t, \qquad \text{and} \qquad g(x_t, t) = g(t) := s_t\sqrt{2\dot{\sigma}_t\sigma_t}. \tag{3}$$

Solving the SDE in Eq. (2) forward in time with an initial condition $x_0$ leads to the Gaussian perturbation kernel $p(x_t|x_0) = \mathcal{N}(x_t; s_t x_0, s_t^2 \sigma_t^2 \mathbf{I})$. Integrating the kernel over the data distribution $p_0(x_0) = p_{\text{data}}$, we obtain the marginal distribution $p_t(x_t)$ at any $t$. As such, one may prescribe the profiles of $s_t$ and $\sigma_t$ so that $p_0 = p_{\text{data}}$ (with $s_0 = 1, \sigma_0 = 0$), and more importantly

$$p_T(x_T) \approx \mathcal{N}(x_T; 0, s_T^2 \sigma_T^2 \mathbf{I}), \tag{4}$$

i.e., the distribution at the terminal time $T$ becomes indistinguishable from an isotropic, zero-mean Gaussian. To sample from $p_{\text{data}}$, we utilize the fact that the reverse-time SDE

$$dx_t = \left[f(t)x_t - g(t)^2 \nabla_{x_t} \log p_t(x_t)\right] dt + g(t)dW_t, \tag{5}$$

has the same marginals as Eq. (2). Thus, by solving Eq. (5) backwards using Eq. (4) as the final condition at time $T$, we obtain samples from $p_{\text{data}}$ at $t = 0$.

Therefore, the problem is reduced to estimating the *score function* $\nabla_{x_t} \log p_t(x_t)$ resulting from $p_{\text{data}}$ and the prescribed diffusion schedule $(s_t, \sigma_t)$. We adopt the denoising formulation in [45] and learn a neural network $D_\theta(x_0 + \varepsilon_t, \sigma_t)$, where $\theta$ denotes the network parameters. The learning seeks to minimize the $L_2$-error in predicting the true sample $x_0$ given a noise level $\sigma_t$ and the sample noised with $\varepsilon_t = \sigma_t \varepsilon$ where $\varepsilon$ is drawn from a standard Gaussian. The score can then be readily obtained from the denoiser $D_\theta$ via the asymptotic relation (i.e., Tweedie's formula [29])

$$\nabla_{x_t} \log p_t(x_t) \approx \frac{D_\theta(\hat{x}_t, \sigma_t) - \hat{x}_t}{s_t \sigma_t^2}, \qquad \hat{x}_t = x_t/s_t. \tag{6}$$

### 3.2 *A posteriori* conditioning via post-processed denoiser

We seek to super-resolve a low-resolution snapshot $\bar{y}' \in \mathcal{Y}'$ to a high-resolution one by leveraging the high-resolution prior modeled by the diffusion model introduced above. Abstractly, our goal is to sample from $p(x_0|E'_{\bar{y}'})$, where $E'_{\bar{y}'} = \{x_0 : C'x_0 = \bar{y}'\}$. Following [30], this may be approximated by modifying the learned denoiser $D_\theta$ at *inference time* (see Appendix A for more details):

$$\tilde{D}_\theta(\hat{x}_t, \sigma_t) = (C')^\dagger \bar{y}' + (I - VV^T)\left[D_\theta(\hat{x}_t, \sigma_t) - \alpha\nabla_{\hat{x}_t}\|C'D_\theta(\hat{x}_t, \sigma_t) - \bar{y}'\|^2\right], \tag{7}$$

where $(C')^\dagger = V\Sigma^{-1}U^T$ is the pseudo-inverse of $C'$ based on its singular value decomposition (SVD) $C' = U\Sigma V^T$, and $\alpha$ is a hyperparameter that is empirically tuned. The $\tilde{D}_\theta$ defined in Eq. (7) directly replaces $D_\theta$ in Eq. (6) to construct a conditional score function $\nabla_{x_t} \log p_t(x_t | E'_{\bar{y}'})$ that facilitates the sampling of $p(x_0 | E'_{\bar{y}'})$ using the reverse-time SDE in Eq. (5).

### 3.3 Debiasing via optimal transport

In order to upsample a biased low-resolution data $\overline{y} \in \mathcal{Y}$, we first seek to find a mapping $T$ such that $\overline{y}' = T(\overline{y}) \in \mathcal{Y}'$ is a representative sample from the distribution of unbiased low-resolution data. Among the infinitely many maps that satisfy this condition, the framework of optimal transport (OT) selects a map by minimizing an integrated transportation distance based on the cost function $c \colon \mathcal{Y} \times \mathcal{Y}' \to \mathbb{R}^+$. The function $c(y, y')$ defines the cost of moving one unit of probability mass from $y'$ to $y$. By treating $Y, Y'$ as random variables on $\mathcal{Y}, \mathcal{Y}'$ with measures $\mu_Y, \mu_{Y'}$, respectively, the OT map is given by the solution to the Monge problem

$$\min_T \left\{ \int c(y, T(y)) d\mu_Y(y) : T_\sharp \mu_Y = \mu_{Y'} \right\}. \tag{8}$$

In practice, directly solving the Monge problem is hard and may not even admit a solution [80]. One common relaxation of Eq. (8) is to seek a joint distribution, known as a coupling or transport plan, which relates the underlying random variables [80]. A valid plan is a probability measure $\gamma$ on $\mathcal{Y} \times \mathcal{Y}'$ with marginals $\mu_Y$ and $\mu_{Y'}$. To efficiently estimate the plan when the $c$ is the quadratic cost (i.e., $c(y, y') = \frac{1}{2}\|y - y'\|^2$), we solve the entropy regularized problem

$$\inf_{\gamma \in \Pi(\mu_Y, \mu_{Y'})} \int \frac{1}{2}\|y - y'\|^2 d\gamma(y, y') + \epsilon D_{\mathrm{KL}}(\gamma \| \mu_{Y'} \otimes \mu_Y), \tag{9}$$

where $D_{\mathrm{KL}}$ denotes the KL divergence, and $\epsilon > 0$ is a small regularization parameter, using the Sinkhorn's algorithm [21], which leverages the structure of the optimal plan to solve Eq. (9) with small runtime complexity [3]. The solution to Eq. (9) is the transport plan $\gamma_\epsilon \in \Pi(\mu, \nu)$ given by

$$\gamma_\epsilon(y, y') = \exp\left( (f_\epsilon(y) + g_\epsilon(y') - \frac{1}{2}\|y - y'\|^2)/\epsilon \right) d\mu_Y(y) d\mu_{Y'}(y'), \tag{10}$$

in terms of potential functions $f_\epsilon, g_\epsilon$ that are chosen to satisfy the marginal constraints. After finding these potentials, we can approximate the transport map using the barycentric projection $T_\gamma(y) = \mathbb{E}_\gamma[Y'|Y = y]$, for a plan $\gamma \in \Pi(\mu_Y, \mu_{Y'})$ [2]. For the plan in Eq. (10), the map is given by

$$T_{\gamma_\epsilon}(y) = \frac{\int y' e^{(g_\epsilon(y') - \frac{1}{2}\|y - y'\|^2)/\epsilon} d\mu_Y(y')}{\int e^{(g_\epsilon(y') - \frac{1}{2}\|y - y'\|^2)/\epsilon} d\mu_Y(y')}. \tag{11}$$

In this work, we estimate the potential functions $f_\epsilon, g_\epsilon$ from samples, i.e., empirical approximations of the measures $\mu_Y, \mu_{Y'}$. Plugging in the estimated potentials in Eq. (11) defines an approximate transport map to push forward samples of $\mu_Y$ to $\mu_{Y'}$. More details on the estimation of the OT map are provided in Appendix H.

A core advantage of this methodology is that it provides us with the flexibility of changing the cost function $c$ in Eq. (8), and embed it with structural biases that one wishes to preserve in the push-forward distribution. Such direction is left for future work.

## 4 Numerical experiments

### 4.1 Data and setup

We showcase the efficacy and performance of the proposed approach on one- and two-dimensional fluid flow problems that are representative of the core difficulties present in numerical simulations of weather and climate. We consider the one-dimensional Kuramoto-Sivashinski (KS) equation and the two-dimensional Navier-Stokes (NS) equation under Kolmogorov forcing (details in Appendix F) in periodic domains. The low-fidelity (LF), low-resolution (LR) data ($\mathcal{Y}$ in Fig. 1(b)) is generated using a finite volume discretization in space [51] and a fractional discretization in time, while the high-fidelity

(HF), high-resolution (HR) data ($\mathcal{X}$ in Fig. 1(b)) is simulated using a spectral discretization in space with an implicit-explicit scheme in time. Both schemes are implemented with `jax-cfd` and its finite-volume and spectral toolboxes [28, 48] respectively. After generating the HF data in HR, we run the LF solver using a spatial discretization that is $8\times$ coarser (in each dimension) with permissible time steps. For NS, we additionally create a $16\times$ coarser LFLR dataset by further downsampling by a factor of two the $8\times$ LFLR data. See Appendix F for further details.

For both systems, the datasets consist of long trajectories generated with random initial conditions[2], which are sufficiently downsampled in time to ensure that consecutive samples are decorrelated. We stress once more that even when the grids and time stamps of both methods are aligned, there is *no pointwise correspondence* between elements of $\mathcal{X}$ and $\mathcal{Y}$. This arises from the different modeling biases inherent to the LF and HF solvers, which inevitably disrupt any short-term correspondence over the long time horizon in a strongly nonlinear dynamical setting.

Finally, we create the intermediate space $\mathcal{Y}'$ in Fig. 1(b) by downsampling the HFHR data with a simple selection mask[3] (i.e., the map $C'$). This creates the new HFLR dataset $\mathcal{Y}'$ with the same resolution as $\mathcal{Y}$, but with the low-frequency bias structure of $\mathcal{X}$ induced by the push-forward of $C'$.

**Baselines and definitions.** We define the following ablating variants of our proposed method

- Unconditional diffusion sampling (*UncondDfn*).
- Diffusion sampling conditioned on LFLR data without OT correction (*Raw cDfn*).
- [*Main*] Diffusion sampling conditioned on OT-corrected (HFLR) data (*OT+cDfn*).

We additionally consider the following baselines to benchmark our method:

- Cubic interpolation approximating HR target using local third-order splines (*Cubic*).
- Vision transformer (*ViT*) [27] based deterministic super-resolution model.
- Bias correction and statistical disaggregation (*BCSD*), involving upsampling with cubic interpolation, followed by a quantile-matching debiasing step.
- CycleGAN, which is adapted from [86] to enable learning transformations between spaces of different dimensions (*cycGAN*).
- ClimAlign (adapted from [32]), in which the input is upsampled using cubic interpolation, and the debiasing step is performed using AlignFlow [34] (*ClimAlign*).

The first two baselines require paired data and, therefore, learn the upsampling map $\mathcal{Y}' \to \mathcal{X}$ (i.e., HFLR to HFHR) and are composed with OT debiasing as factorized baselines. BCSD is a common approach used in the climate literature. The last two baselines present end-to-end alternatives and are trained directly on unpaired LFLR and HFHR samples. Further information about the implemented baselines can be found in Appendix D.

**OT training.** To learn the transport map in Eq. (11), we solve the entropic OT problem in Eq. (9) with $\epsilon = 0.001$ using a Sinkhorn [21] iteration with Anderson acceleration and parallel updates. We use $90,000$ i.i.d. samples of $Y \in \mathcal{Y}$ and $Y' \in \mathcal{Y}'$, and perform 5000 iterations. Implementations are based on the `ott-jax` library [22].

**Denoiser training and conditional sampling.** The denoiser $D_\theta$ is parametrized with a standard U-Net architecture similar to the one used in [67]. We additionally incorporate the preconditioning technique proposed in [45]. For $s_t$ and $\sigma_t$ schedules, we employ the variance-preserving (VP) scheme originally introduced in [73]. Furthermore, we adopt a data augmentation procedure to increase the effective training data size by taking advantage of the translation symmetries in the studied systems.

Samples are generated by solving the SDE based on the post-processed denoiser $\tilde{D}_\theta$ using the Euler-Maruyama scheme with exponential time steps, i.e., $\{t_i\}$ is set such that $\sigma(t_i) = \sigma_{\max}(\sigma_{\min}/\sigma_{\max})^{i/N}$ for $i = \{0, ..., N\}$. The number of steps used, $N$, vary between systems and downscaling factors. More details regarding denoiser training and sampling are included in Appendix B.

**Metrics.** To quantitatively assess the quality of the resulting snapshots we compare a number of physical and statistical properties of the snapshots: (i) the energy spectrum, which measures the

---

[2]The presence of global attractors in both systems renders the exact initial conditions unimportant. It also guarantees sufficient coverage of the target distributions sampling from long trajectories.

[3]It is worth noting that careful consideration should be given to the choice of $C'$ to avoid introducing aliasing, as this can potentially make the downscaling task more challenging.

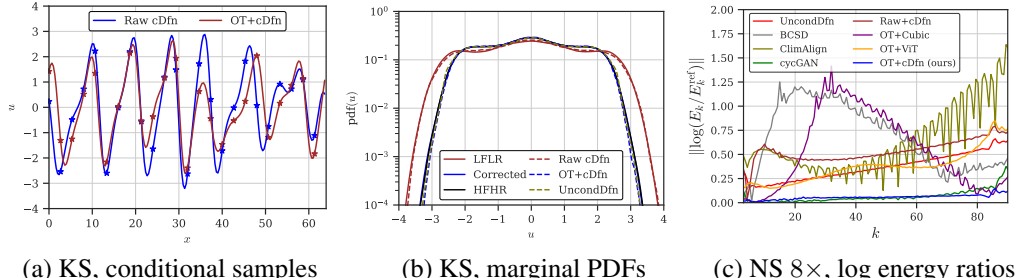

| (a) KS, conditional samples | (b) KS, marginal PDFs | (c) NS $8\times$, log energy ratios |

Figure 2: (a) KS samples generated with diffusion model conditioned on LR information with and without OT correction applied, (b) empirical probability density function for relevant LR and HR samples in KS and (c) mode-wise log energy ratios with respect to the true samples (Eq. (13) without weighted sum) at $8\times$ downscaling for NS.

Table 1: Metrics of the LFLR source and OT-corrected samples for KS and NS. The precise metric definitions are provided in Appendix C.

| | KS $8\times$ | | NS $8\times$ | | NS $16\times$ | |
|---|---|---|---|---|---|---|
| **Metric** | LFLR | OT-corrected | LFLR | OT-corrected | LFLR | OT-corrected |
| covRMSE $\downarrow$ | 0.343 | **0.081** | 0.458 | **0.083** | 0.477 | **0.079** |
| MELRu $\downarrow$ | 0.201 | **0.020** | 1.254 | **0.013** | 0.600 | **0.016** |
| MELRw $\downarrow$ | 0.144 | **0.020** | 0.196 | **0.026** | 0.200 | **0.025** |
| KLD $\downarrow$ | 1.464 | **0.018** | 29.30 | **0.033** | 12.26 | **0.017** |

energy in each Fourier mode and thereby providing insights into the similarity between the generated and reference samples, (ii) a spatial covariance metric, which characterizes the spatial correlations within the snapshots, (iii) the KL-divergence (KLD) of the kernel density estimation for each point, which serves as a measure for the local structures (iv) the maximum mean discrepancy (MMD), and (v) the empirical Wasserstein-1 metric (Wass1). We present (i) below and leave the rest described in Appendix C as they are commonly used in the context of probabilistic modeling.

The energy spectrum is defined[4] as

$$E(k) = \sum_{|\underline{k}|=k} |\hat{u}(\underline{k})|^2 = \sum_{|\underline{k}|=k} \left| \sum_i u(x_i) \exp(-j2\pi \underline{k} \cdot x_i/L) \right|^2 \tag{12}$$

where $u$ is a snapshot system state, and $k$ is the magnitude of the wave-number (wave-vector in 2D) $\underline{k}$. To assess the overall consistency of the spectrum between the generated and reference samples using a single scalar measure, we consider the mean energy log ratio (MELR):

$$\text{MELR} = \sum_k w_k \left| \log \left( E_{\text{pred}}(k)/E_{\text{ref}}(k) \right) \right|, \tag{13}$$

where $w_k$ represents the weight assigned to each $k$. We further define $w_k^{\text{unweighted}} = 1/\text{card}(k)$ and $w_k^{\text{weighted}} = E_{\text{ref}}(k)/\sum_k E_{\text{ref}}(k)$. The latter skews more towards high-energy/low-frequency modes.

## 4.2 Main results

**Effective debiasing via optimal transport.** Table 1 shows that the OT map effectively corrects the statistical biases in the LF snapshots for all three experiments considered. Significant improvements are observed across all metrics, demonstrating that the OT map approximately achieves $C'_\sharp \mu_X \approx T_\sharp \mu_Y$ as elaborated in §3 (extra comparisons are included in Appendix H).

Indeed, the OT correction proves crucial for the success of our subsequent conditional sampling procedure: the unconditional diffusion samples may not have the correct energy spectrum (see

---

[4]This definition is applied to each sample and averaged to obtain the metric (same for MELR below).

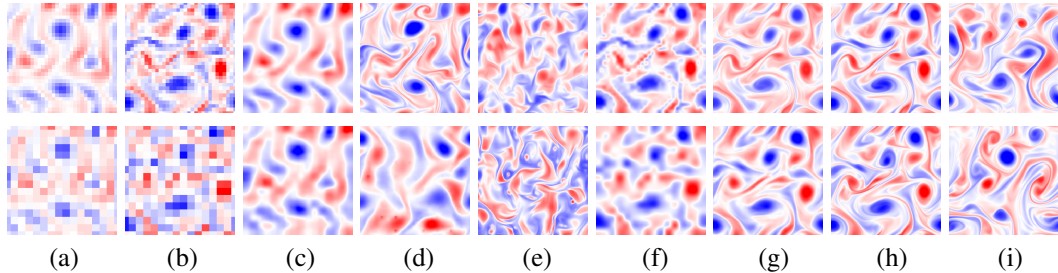

|     |     |     |     |     |     |     |     |     |
| (a) | (b) | (c) | (d) | (e) | (f) | (g) | (h) | (i) |

Figure 3: Example showing the vorticity field of samples debiased and super-resolved using different techniques at $8\times$ (top row) and $16\times$ (bottom row) downscaling factors. From left to right: **(a)** LR snapshots produced by the **low-fidelity solver** (input $\bar{y}$ of Alg. 1), **(b) OT-corrected** snapshots ($\bar{y}'$ in line 1 of Alg. 1), **(c) BCSD** applied to LR snapshots, **(d)** snapshots downscaled with **cycle-GAN** directly from LR snapshots, **(e) ClimAlign** applied to LR snapshots, **(f) cubic interpolation** of the OT-corrected snapshots, **(g)** deterministic upsample of the OT-corrected snapshots with **ViT**, **(h) diffusion sample conditioned on the OT-corrected snapshots** (output $\bar{x}$ in Alg. 1, ours), and **(i)** two **true HR samples** in the training data with the closest Euclidean distance to the OT-corrected generated sample. The $16\times$ source is the same as the $8\times$ source but further downsampled by a factor of two. OT maps are computed independently between resolutions.

*UncondDfn* in Fig. 2(c), i.e. suffering from *color shifts* - a known problem for score-based diffusion models [72, 15]. The conditioning on OT corrected data serves as a sparse anchor which draws the diffusion trajectories to the correct statistics at sampling time. In fact, when conditioned on uncorrected data, the bias effectively pollutes the statistics of the samples (*Raw cDfn* in Table 2). Fig. 2(b) shows that the same pollution is present for the KS case, despite the unconditional sampler being unbiased.

In Appendix E, we present additional ablation studies that demonstrate the importance of OT correction in the factorized benchmarks.

**Comparison vs. factorized alternatives.** Fig. 3 displays NS samples generated by all benchmarked methods. Qualitatively, our method is able to provide highly realistic small-scale features. In comparison, we observe that *Cubic* expectedly yields the lowest quality results; the deterministic *ViT* produces samples with color shift and excessive smoothing, especially at $16\times$ downscaling factor.

Quantitatively, our method outperforms all competitors in terms of MELR and KLD metrics in the NS tasks, while demonstrating consistently good performance in both $8\times$ and $16\times$ downscaling, despite the lack of recognizable features in the uncorrected LR data (Fig. 3(a) bottom) in the latter case. Other baselines, on the other hand, experience a significant performance drop. This showcases the value of having an unconditional prior to rely on when the conditioning provides limited information.

**Comparison vs. end-to-end downscaling.** Although the *cycGAN* baseline is capable of generating high-quality samples at $8\times$ downscaling (albeit with some smoothing) reflecting competitive metrics, we encountered persistent stability issues during training, particularly in the $16\times$ downscaling case.

**Diffusion samples exhibit ample variability.** Due to the probabilistic nature of our approach, we can observe from Table 2 that the OT-conditioned diffusion model provides some variability in the downscaling task, which increases when the downscaling factor increases. This variability provides a measure of uncertainty quantification in the generated snapshots as a result of the consistent formulation of our approach on probability spaces.

## 5   Conclusion

We introduced a two-stage probabilistic framework for the statistical downscaling problem. The framework performs a debiasing step to correct the low-frequency statistics, followed by an up-sampling step using a conditional diffusion model. We demonstrate that when applied to idealized physical fluids, our method provides high-resolution samples whose statistics are physically correct, even when there is a mismatch in the low-frequency energy spectra between the low- and high-

Table 2: Evaluation of downscaling methods for NS. The best metric values are highlighted **in bold**. Precise metric definitions (except MELR, given by Eq. (13)) are included in Appendix C.

| Model | Var | covRMSE↓ | MELRu↓ | MELRw↓ | KLD↓ | Wass1↓ | MMD↓ |
|---|---|---|---|---|---|---|---|
| **8× downscale** | | | | | | | |
| BCSD | 0 | 0.31 | 0.67 | 0.25 | 2.19 | **0.23** | 0.10 |
| cycGAN | 0 | 0.15 | 0.08 | 0.05 | 1.62 | 0.32 | 0.08 |
| ClimAlign | 0 | 2.19 | 0.64 | 0.45 | 64.37 | 2.77 | 0.53 |
| Raw+cDfn | 0.27 | 0.46 | 0.79 | 0.37 | 73.16 | 1.04 | 0.42 |
| OT+Cubic | 0 | **0.12** | 0.52 | 0.06 | 1.46 | 0.42 | 0.10 |
| OT+ViT | 0 | 0.43 | 0.38 | 0.18 | 1.72 | 1.11 | 0.31 |
| (ours) OT+cDfn | 0.36 | **0.12** | **0.06** | **0.02** | **1.40** | 0.26 | **0.07** |
| **16× downscale** | | | | | | | |
| BCSD | 0 | 0.34 | 0.67 | 0.25 | 2.17 | **0.21** | 0.11 |
| cycGAN | 0 | 0.32 | 1.14 | 0.28 | 2.05 | 0.48 | 0.13 |
| ClimAlign | 0 | 2.53 | 0.81 | 0.50 | 77.51 | 3.15 | 0.55 |
| Raw+cDfn | 1.07 | 0.46 | 0.54 | 0.30 | 93.87 | 0.99 | 0.39 |
| OT+Cubic | 0 | 0.25 | 0.55 | 0.13 | 7.30 | 0.85 | 0.20 |
| OT+ViT | 0 | 0.14 | 1.38 | 0.09 | 1.67 | 0.32 | **0.07** |
| (ours) OT+cDfn | 1.56 | **0.12** | **0.05** | **0.02** | **0.83** | 0.29 | **0.07** |

resolution data distributions. We have shown that our method is competitive and outperforms several commonly used alternative methods.

Future work will consider fine-tuning transport maps by adapting the map to the goal of conditional sampling, and introducing physically-motivated cost functions in the debiasing map. Moreover, we will address current limitations of the methodology, such as the high-computational complexity of learning OT-maps that scales quadratically with the size of the training set, and investigate the model's robustness to added noise in the collected samples as is found in weather and climate datasets. We will also further develop this methodology to cover other downscaling setups such as perfect prognosis [57] and spatio-temporal downscaling.

## Broader impact

Statistical downscaling is important to weather and climate modeling. In this work, we propose a new method for improving the accuracy of high-resolution forecasts (on which risk assessment would be made) from low resolution climate modeling. Weather and climate research and other scientific communities in computational fluid dynamics will benefit from this work for its potential to reduce computational costs. We do not believe this research will disadvantage anyone.

## Acknowledgments

The authors would like to sincerely thank Toby Bischoff, Katherine Deck, Nikola Kovachki, Andrew Stuart and Hongkai Zheng for many insightful and inspiring discussions that were vital to this work. RB gratefully acknowledges support from the Air Force Office of Scientific Research MURI on "Machine Learning and Physics-Based Modeling and Simulation" (award FA9550-20-1-0358), and a Department of Defense (DoD) Vannevar Bush Faculty Fellowship (award N00014-22-1-2790).

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
