## A    Constrained sampling via post-processed denoiser

In this section, we provide more details on the apparatus necessary to perform *a posteriori* conditional sampling in the presence of a linear constraint.

Eq. (6) suggests that the SDE drift corresponding to the score may be broken down into 3 steps:

1. The denoiser output $D_\theta(\hat{x}_t, \sigma_t)$ provides a *target sample* that is estimated to be a member of the true data distribution;
2. The current state $\hat{x}_t$ is *nudged* towards this target[5];
3. Appropriate rescaling is applied based on the scale and noise levels ($s_t$ and $\sigma_t$) prescribed for the current diffusion time $t$.

For conditional sampling, consider imposing a linear constraint $Cx_0 = y$, where $C \in \mathbb{R}^{d \times d_c}$ and $y \in \mathbb{R}^{d_c}$. Decomposing $C$ in terms of its singular value decomposition (SVD), $C = U\Sigma V^T$, leads to

$$V^T x_0 = \Sigma^{-1} U^T y := \tilde{y}. \tag{14}$$

Note that this constraint can be easily embedded in the target by replacing the corresponding components of $D_\theta(\hat{x}_t, \sigma_t)$ in this subspace with the constrained value, yielding a post-processed target

$$D_{\theta,\text{cons}}(\hat{x}_t, \sigma_t) = V\tilde{y} + (I - VV^T)D_\theta(\hat{x}_t, \sigma_t). \tag{15}$$

This modification alone *guarantees* that the sample $x_0$ produced by the SDE satisfies the required constraint (up to solver errors), while components in the orthogonal complement of the constraint are guided by the denoiser just as in the unconstrained case.

However, in practice this modification creates a "discontinuity" between the constrained and unconstrained components, leading to erroneous correlations between them in the generated samples. As a remedy, we introduce an additional correction in the unconstrained subspace:

$$\tilde{D}_{\theta,\text{cons}}(\hat{x}_t, \sigma_t) = D_{\theta,\text{cons}}(\hat{x}_t, \sigma_t) - \alpha(I - VV^T)\nabla_{\hat{x}_t} L(D_\theta(\hat{x}_t, \sigma_t)), \tag{16}$$

where

$$L(\hat{x}_t) = \|CD_\theta(\hat{x}_t, \sigma_t) - y\|^2 \tag{17}$$

is a loss function measuring how well the denoiser output conforms to the imposed constraint. This is the post-processed denoiser function Eq. (7) in the main text. The extra correction term effectively induces a gradient descent roughly in the form

$$\dot{\hat{x}}_t = -\hat{\alpha}\nabla_{\hat{x}_t} L(D_\theta(\hat{x}_t, \sigma_t)) \tag{18}$$

with respect to loss function $L$ *in the dynamics of the unconstrained components*. $\hat{\alpha}$ is a positive "learning rate" that is determined empirically such that the loss value reduces adequately close to zero by the conclusion of the denoising process. Besides the $1/s_t\sigma_t^2$ scaling bestowed by the diffusion process, it also depends on the scaling of the constraint matrix $C$, and in turn directly influences the permissible solver discretization during sampling. Thus it needs to be tuned empirically.

Substituting $\tilde{D}_{\theta,\text{cons}}$ for $D_\theta(\hat{x}_t, \sigma_t)$ in Eq. (6) results in the conditional score

$$\nabla_{x_t} \log p_t(x_t | E_y) = \frac{\tilde{D}_{\theta,\text{cons}}(\hat{x}_t, \sigma_t) - \hat{x}_t}{s_t\sigma_t^2}. \tag{19}$$

Note that the same re-scale $1/s_t\sigma_t^2$ is applied as before.

**Remark 1.** The correction in Eq. (16) is equivalent to imposing a Gaussian likelihood on $x_0$ (and thus the linearly transformed $Cx_0$) given $x_t$. To see this, first note that that applying Bayes' rule to the conditional score function results in

$$\nabla_{x_t} \log p_t(x_t | E_y) = \nabla_{x_t} \log p_t(x_t) + \nabla_{x_t} \log p(Cx_0 = y | x_t), \tag{20}$$

where the probability in the second term may be viewed as a likelihood function for $Cx_0$.

---

[5]In the same way that $\dot{a} = -\beta(a_0 - a)$ results in $a \to a_0$ as $t \to -\infty$ for any $\beta > 0$

Next, substituting Eq. (16) into Eq. (19) yields

$$\nabla_{x_t} \log p_t(x_t|E_y) = \frac{D_{\theta,\text{cons}}(\hat{x}_t, \sigma_t) - \hat{x}_t}{s_t \sigma_t^2} + (I - VV^T)\frac{-\alpha}{s_t \sigma_t^2}\nabla_{\hat{x}_t}\|CD_\theta(\hat{x}_t, \sigma_t) - y\|^2 \quad (21)$$

where the second term (without the projection) may be further rewritten as

$$-\frac{\alpha}{s_t \sigma_t^2}\nabla_{\hat{x}_t}\|CD_\theta(\hat{x}_t, \sigma_t) - y\|^2 = -\frac{\alpha}{\sigma_t^2}\nabla_{x_t}\|CD_\theta(\hat{x}_t, \sigma_t) - y\|^2 \quad (22)$$

$$= \nabla_{x_t} \log\left(\exp\left(-\frac{2\alpha}{2\sigma_t^2}\|CD_\theta(\hat{x}_t, \sigma_t) - y\|^2\right)\right) \quad (23)$$

$$= \nabla_{x_t} \log \mathcal{N}\left(y; CD_\theta(\hat{x}_t, \sigma_t), \frac{\sigma_t^2}{2\alpha}I\right). \quad (24)$$

In other words, the correction is equivalent to imposing an isotropic Gaussian likelihood model for $y$ with mean $CD_\theta(\hat{x}_t, \sigma_t)$ and variance $\sigma_t^2/2\alpha$. It is worth noting that both the mean and variance here have direct correspondence to estimations of statistical moments using Tweedie's formulas [29]:

$$\mathbb{E}[x_0|x_t] = D_\theta(\hat{x}_t, \sigma_t) \qquad \text{and} \qquad \text{Cov}[x_0|x_t] = \sigma_t^2 \nabla_{\hat{x}_t} D_\theta(\hat{x}_t, t), \quad (25)$$

with an additional approximation for the (linearly transformed) covariance

$$C\nabla_{\hat{x}_t} D_\theta(\hat{x}_t, t)C^T \approx \frac{1}{2\alpha}I, \quad (26)$$

which is expensive to evaluate in practice.

Lastly, it is important to note that the true likelihood $p(x_0|x_t) \propto p(x_t|x_0)p(x_0)$ is in general not Gaussian unless the target data distribution $p(x_0)$ is itself Gaussian. However, the Gaussian assumption is good at early stages of denoising ($t \gg 0$) when the signal-to-noise ratio (SNR) is low. Later on, the true likelihood becomes closer to a $\delta$-distribution as $\sigma_t \to 0$, and the denoising is in turn dictated by the mean.

**Remark 2.** The post-processing presented in this section is similar to [17], who propose to apply a correction proportional to $\nabla_{\hat{x}_t}\|CD_\theta(\hat{x}_t, \sigma_t) - y\|^2$ directly to the score function. The main difference is the lack of the additional scaling $\sigma_t^2$ that adapts to the changing noise levels in the denoise process. In practice, we found that including this scaling contributes greatly to the numerical stability and efficiency of continuous-time sampling.

## B  Diffusion model details

### B.1  Training

The training of our denoiser-based diffusion models largely follows the methodology proposed in [45]. In this section, we present the most relevant components for completeness and better reproducibility.

The variance-preserving (VP) schedule sets the forward SDE parameters:

$$\sigma_t = \sqrt{e^{\frac{1}{2}\beta_d t^2 + \beta_{\min} t} - 1}, \qquad s_t = 1/\sqrt{e^{\frac{1}{2}\beta_d t^2 + \beta_{\min} t}} = 1/\sqrt{\sigma_t^2 + 1}, \quad (27)$$

with $\beta_b = 19.9$, $\beta_{\min} = 0.1$ and time $t$ going from 0 to 1.

The loss function for training the denoiser $D_\theta$ reads as

$$L(\theta) = \sum_i \lambda(\sigma_{t_i})\|D_\theta(x_{0,i} + \sigma_{t_i}\varepsilon, \sigma_{t_i}) - x_{0,i}\|^2 \quad (28)$$

over a batch $\{x_{0,i}, t_i\}$ of size $N_{\text{batch}}$ indexed by $i$, with $x_{0,i} \sim p_{\text{data}}$ and $\varepsilon \sim \mathcal{N}(0, I)$. The times $\{t_i\}$ are selected such that

$$t_{i+1} - t_i = \Delta t, \quad t_0 \sim \mathcal{U}[\epsilon_t, \epsilon_t + \Delta t], \quad \Delta t = \frac{1 - \epsilon_t}{N_{\text{batch}}}. \quad (29)$$

That is, the times are evenly spaced out in $[\epsilon_t, 1]$ with interval $\Delta t$ given a random starting point $t_0$. $\epsilon_t = 10^{-3}$ is the minimum time set to prevent numerical blow-up. $\lambda$ is the weight assigned to the loss at noise level $\sigma_{t_i}$, which is given by

$$\lambda(\sigma_{t_i}) = (\sigma_{t_i}^2 + \sigma_{\text{data}}^2)/(\sigma_{t_i}\sigma_{\text{data}})^2, \tag{30}$$

where $\sigma_{\text{data}}^2$ is the data variance.

We further adopt the preconditioned denoiser ansatz

$$D_\theta(x, \sigma) = \frac{\sigma_{\text{data}}^2}{\sigma_{\text{data}}^2 + \sigma^2}x + \frac{\sigma_{\text{data}}\sigma}{\sqrt{\sigma_{\text{data}}^2 + \sigma^2}}F_\theta\left(\frac{x}{\sqrt{\sigma_{\text{data}}^2 + \sigma^2}}, \frac{1}{4}\log(\sigma)\right), \tag{31}$$

where $F_\theta$ is the raw U-Net model. This ansatz ensures that the training inputs and targets of $F_\theta$ both roughly have unit variance, and the approximation errors in $F_\theta$ are minimally amplified in $D_\theta$ across all noise levels (see Appendix B.6 in [45]).

**Data augmentation** Both KS and NS exhibit translation symmetry (only in the $x$-direction for NS due to the $y$-dependence of the Kolmogorov forcing, see section F), meaning that $u(x + a)$ (or $u(x + a, y)$ for NS) is automatically a valid sample for any constant scalar $a$ provided that $u(x)$ (or $u(x, y)$ for NS) is a valid sample. We leverage this property, as well as the fact that both systems are subject to periodic boundary conditions, to augment our dataset by applying a `numpy.roll` operation with a random shift.

## B.2 Sampling

The reverse SDE in Eq. (5) used for sampling may be rewritten in terms of denoiser $D_\theta$ as

$$dx_t = \left[\left(\frac{\dot{\sigma}_t}{\sigma_t} + \frac{2\dot{s}_t}{s_t}\right)x_t - \frac{2s_t\dot{\sigma}_t}{\sigma_t}D_\theta\left(\frac{x_t}{s_t}, \sigma_t\right)\right]dt + s_t\sqrt{2\dot{\sigma}_t\sigma_t}\,dW_t. \tag{32}$$

Parts of the drift term inside the squared brackets are inversely proportional to $\sigma_t$ and hence quickly rises in magnitude as $\sigma_t \to 0$. This means that the dynamics becomes *stiffer* as $t \to 0$, necessitating the use of progressively finer time steps during denoising. As stated in §4.1 of the main text, for this very reason, we employ an exponential profile with non-uniform time steps proportional to $\sigma_t$.

Similarly, the stiffness of the dynamics also increases with the conditioning strength $\alpha$ in the post-processed denoiser $\tilde{D}_\theta$ in Eq. (7). Therefore, for each conditional sampling setting (downscaling factor and $C'$ map), we use an *ad hoc* number of steps, as determined empirically from a grid search (section E.1).

## C Metrics

### C.1 Definitions

In this section, we present the definitions of additional metrics that are used for the comparisons in §4.2. The energy-based metrics are already defined in Eq. (12) and Eq. (13) of the main text.

**Relative root mean squared error (RMSE)** is defined as

$$\text{RMSE} = \frac{1}{N}\sum_{n=1}^{N}\frac{\|z_{\text{pred},n} - z_{\text{ref},n}\|_2}{\|z_{\text{pred},n}\|_2}, \tag{33}$$

where the predicted and reference quantities $z_{\text{pred},n}$ and $z_{\text{ref},n}$ are computed over an evaluation batch (of size $N$, indexed by $n$). The *constraint RMSE* corresponds to this metric evaluated on the conditioned pixels of the generated samples (predicted) and the conditioned values $\bar{y}'$ (reference). It provides a measure for how well the generated conditional samples satisfy the imposed constraint.

**Covariance RMSE (covRMSE)** referenced in Table 1 corresponds to computing Eq. (33) between the (empirical) covariance matrices of the generated and reference samples given by

$$\text{Cov}(u) = \frac{1}{N}\sum_{n=1}^{N}(u_n - \bar{u})(u_n - \bar{u})^T, \quad \bar{u} = \frac{1}{N}\sum_{n=1}^{N}u_n, \tag{34}$$

Table 3: Number of samples used for evaluation. For sampling runs which are deterministic or unconditional in nature, the number of evaluation samples is equal to the number of OT samples (rather than the number of conditions) to ensure convergence in statistics.

| | OT | Conditional sampling | |
|---|---|---|---|
| **System** | samples | conditions | samples per condition |
| KS | 15360 | 512 | 128 |
| NS | 10240 | 128 | 128 |

where $u_n$ are realizations of the multi-dimensional random variable $U$. For KS, we compute the covariance along the full domain by treating each pixel as a distinct dimension of the random variable. For NS, we leverage the translation invariance in the system to compute the covariance on slices with fixed $x$-coordinate (i.e., dimensions are indexed by the $y$-coordinate). Lastly, since we are dealing with matrices, the norm involved in Eq. (33) is taken to be the Frobenious norm.

**Kernel-density-estimated Kullback-Leibler divergence (KLD)** computes the KL divergence using 1-dimensional marginal kernel density estimations (KDEs, with the bandwidths selected based on Scott's rule [70]). That is,

$$\text{KLD} = \sum_{m=1}^{d} \int_{-\infty}^{\infty} \tilde{p}_{d,\text{ref}}(\upsilon) \log \left( \frac{\tilde{p}_{d,\text{ref}}(\upsilon)}{\tilde{p}_{d,\text{pred}}(\upsilon)} \right) d\upsilon, \tag{35}$$

where $\tilde{p}_m$ are empirical probability density functions (PDFs) obtained with KDE for a particular dimension $m$ of the samples. The integral is approximated using the trapezoidal rule, and summed over all dimensions for an aggregated measure, as if they were independent.

**Sample variability (Var)** refers to the mean pixel-wise standard deviation in the generated conditional samples given by

$$\text{Var} = \sqrt{\frac{1}{Nd} \sum_n^N \sum_m^d (u_{nm} - \bar{u}_m)^2}, \quad \bar{u}_m = \frac{1}{N} \sum_{n=1}^N u_{nm}, \tag{36}$$

where $\bar{u}_m$ is obtained by averaging the values of dimension $m$ over samples *with the same condition*.

**Mean Maximum Discrepancy (MMD)** is computed using the following empirical estimation

$$\begin{aligned} \text{MMD}^2 = &\frac{1}{N_p(N_p - 1)} \sum_{i,j \neq i} k(z_{\text{pred},i}, z_{\text{pred},j}) - \frac{2}{N_p N_r} \sum_{i,j} k(z_{\text{pred},i}, z_{\text{ref},j}) \\ &+ \frac{1}{N_r(N_r - 1)} \sum_{i,j \neq i} k(z_{\text{ref},i}, z_{\text{ref},j}), \end{aligned} \tag{37}$$

between generated and reference samples $\{z_{\text{pred}}\}$ and $\{z_{\text{ref}}\}$. For $k$ we use a multi-scale Gaussian kernel with bandwidths $[2, 4, 6, 8] \times 256$, which are tuned empirically to the rough scales of the reference distribution.

**Wasserstein-1 metric (Wass1)** is given by

$$\text{Wass1} = \frac{1}{d} \sum_m^d \int \left| \text{CDF}_{\text{pred},d}(z) - \text{CDF}_{\text{ref},d}(z) \right| dz, \tag{38}$$

where the 1-dimensional CDFs are empirically computed with `np.histogram` and averaged across all dimensions. The integral is performed over the range $[-20, 20]$.

**Evaluation setup.** The number of samples used to evaluate the metrics is summarized in Table 3. MELR and KLD metrics are evaluated marginally, i.e., on all conditional samples pooled together.

## C.2 Additional results

Table 4 shows the conditional sampling metrics for KS. Additional energy spectra and log energy ratio calculations for both systems are displayed in Fig. 4.

Table 4: Additional KS conditional sampling metrics.

| Method | Constraint RMSE | Sample Variability | MELR (unweighted) | MELR (weighted) | KLD |
|---|---|---|---|---|---|
| *Raw cDfn* | 0.001 | 0.044 | 0.527 | 0.143 | 10.37 |
| *OT+cDfn* | 0.001 | 0.044 | 0.362 | 0.044 | 1.27 |

Table 5: LR metrics for NS, computed for downsampled outputs of end-to-end baselines (BCSD, cycGAN and ClimAlign). OT is superior in distributional metrics (covRMSE, MELR, MMD) but pays a "price" in terms of pixel-wise similarity represented in the sMAPE metric (between corrected and uncorrected LR snapshots).

|  | OT | BCSD | cycGAN | ClimAlign |
|---|---|---|---|---|
| **8×downscale** | | | | |
| covRMSE | 0.08 | 0.31 | 0.16 | 2.21 |
| MELRu | 0.01 | 0.95 | 0.08 | 0.53 |
| MELRw | 0.03 | 0.13 | 0.04 | 0.54 |
| MMD | 0.04 | 0.06 | 0.06 | 0.61 |
| sMAPE | 0.53 | 0.25 | 0.41 | 0.74 |
| **16×downscale** | | | | |
| covRMSE | 0.08 | 0.35 | 0.33 | 2.50 |
| MELRu | 0.02 | 0.63 | 0.34 | 0.67 |
| MELRw | 0.03 | 0.16 | 0.15 | 0.58 |
| MMD | 0.03 | 0.34 | 0.09 | 0.55 |
| sMAPE | 0.54 | 0.36 | 0.63 | 0.76 |

Table 5 contrasts OT with end-to-end baselines. Since end-to-end baselines directly output HR samples, they are downsampled to LR to enable apple-to-apple comparison. OT achieves the best distributional metrics. Note that this comes seemingly "at the price" of decreased pixel-wise similarity, which may be quantified through the symmetric mean absolute percentage error (sMAPE):

$$\text{sMAPE} = \frac{1}{N} \sum_{n=1}^{N} \frac{|y_n - y'_n|}{(|y_n| + |y'_n|)/2}, \tag{39}$$

where $y_n$ and $y'_n$ denote the LFLR and the downsampled end-to-end baseline outputs. Note that sMAPE more closely embodies the "visual discrepancy" one observes before and after debiasing. We reemphasize that this is a feature inherent to distribution-based debiasing and in fact an intended consequence.

# D  Baselines

## D.1  Cubic interpolation

Cubic interpolation employs a local third-order polynomial for the interpolation process. It builds a local third order polynomial, or a cubic spline, in the form

$$u(x, y) = \sum_{i=0}^{3} \sum_{j=0}^{3} a_{ij} x^i y^j, \tag{40}$$

where the coefficients $a_{ij}$ are usually found using Lagrange polynomials. We use the function `jax.image.resize` to perform the interpolation.

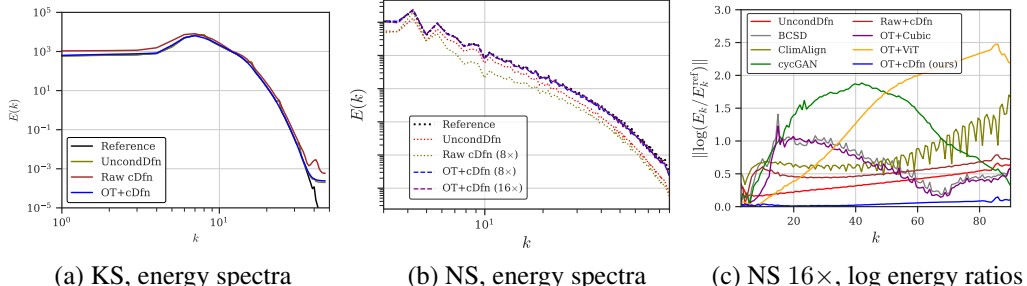

|  (a) KS, energy spectra | (b) NS, energy spectra | (c) NS $16\times$, log energy ratios |

Figure 4: (a) Sample energy spectra (Eq. (12)) comparison in KS, (b) sample energy spectra comparison in NS and (c) mode-wise log energy ratios with respect to the true samples (Eq. (13) without weighted sum) at $16\times$ downscaling for NS.

## D.2 BCSD

Bias correction and statistical disaggregation (BCSD) [56] is a two-stage downscaling procedure. It first implements a cubic interpolation (using `jax.image.resize`), and then performs pixel-wise quantile matching.

For quantile matching, we use the `tensorflow.probability` library. Specifically, for each point in both the interpolated and reference HRHF snapshots, we compute the segments corresponding to 1000 quantiles of each distribution using the `stats.quantiles()` function. At inference time, for each pixel in the interpolated snapshot, we perform the following steps: (i) find the closest segment (out of the 1000 quantiles) that contains the value at the given pixel, (ii) identify the quantile corresponding to that segment, (iii) find the segment corresponding to that quantile in the HRHF data, and finally, (iv) output the middle point in that quantile.

The number of quantiles was chosen to minimize the Wasserstein norm, while reliably computing the quantiles from the $90,000$ samples. Finally, we note that quantile matching is indeed the minimizer of the Wasserstein-1 norm, which in the one dimensional case can be conveniently expressed as the $L^1$ distance between the cumulative distribution functions of the corresponding measures.

## D.3 ViT model

For the deterministic upsampling model, we consider a Vision Transformer (ViT) model similar to several CNN based super-resolution models [26], but with a Transformer core that significantly increases the capacity of the model. Our model follows the standard structure of a ViT. However, it differs in the tokenization step, where instead of using a linear transformation from a patch of the input to a embedding, we employ a single-pixel embedding combined with a series of downsampling blocks. Each downsampling block consists of a sequence of ResNet blocks and a coarsening layer implemented using a strided convolution. This architectural choice draws inspiration from the hierarchical processing of CNNs [26, 86]. After tokenization, the tokens are processed using self-attention blocks following [27]. The outputs of the self-attention blocks are then upsampled using upsampling blocks. Each upsampling block consists of a nearest neighbor upsampling layer, which combines nearest neighbor interpolation and a convolution layer, followed by a sequence of ResNet blocks. We provide below more details on the implementation of each block and the core architecture.

**Embedding**  We use a $1 \times 1$ convolution to implement a pixel-wise embedding whose dimension was tuned in a hyperparameter sweep.

**Downsampling blocks**  The downscaling blocks quadruple the number of channels of the input as the other dimensions are decimated by a factor two (red blocks in Fig. 5). This is achieved with a convolutional layer with a $(2, 2)$ stride, a fixed kernel width (hyperparameter) and periodic (i.e. circular) boundary conditions. After the convolution, we use a sequence of convolutional ResNet layers, with a GeLU activation function and a layer normalization. These convolutional layers also use a fixed kernel width and periodic boundary conditions. The number of downsampling blocks and the number of ResNet blocks inside each layer were empirically tuned.

**Transformer core** Then the output of the downsampling blocks is reshaped into a sequence of tokens, in which a corresponding 2-dimensional embedding is added to account for the underlying geometry in the self-attention blocks. We then perform a sequence of self-attention blocks following the blocks introduced in [27] (blue blocks in Fig. 5). The number of attention heads of each self-attention block is equal to the dimension. The number of transformer blocks is also tuned. The GeLU activation functions is used for the self-attention layers.

**Upsampling blocks** The tokens are reshaped back to their original 2-dimensional topology. Then, a sequence of upsampling layers followed by a handful of ResNet blocks is used to downscale the image (green blocks in Fig. 5) at its target resolution (purple block in Fig. 5). At each upsampling step, the spatial resolution is increased by a factor of two in each dimension, while reducing the number of channel so that the overall information remains constant. As mentioned above, the upsampling is performed using a nearest neighbor interpolation (we repeat the value of the closest neighbor), followed by a linear convolutional layer with a $(3, 3)$ kernel size, and subsequently a series of ResNet blocks similar to those used in the downsampling block.

The number of downsampling and upsampling blocks were chosen to strike a computational balance between the quadratic complexity of the Transformer core on the number of tokens and the quadratic complexity on the width of each token.

We use a simple mean squared error loss given that in this case we have paired data. As we seek to build a upsampling network from $\mathcal{Y}'$ to $\mathcal{X}$, the inputs $y' \in \mathcal{Y}'$ are nothing more than the desired output but downsampled by a factor 8 or 16, or $y' = C'x$ for $x \in \mathcal{X}$. We observed that the network in the first runs were not able to faithfully interpolate the input, i.e., if we denote the network by $\mathcal{N}_\theta$, where $\theta$ corresponds to the set of parameters, then the interpolation should satisfy $C'\mathcal{N}_\theta(y') = (y')$. Therefore, we added a regularization term in the loss weighted by a tunable parameter $\lambda$. In a nutshell, the loss is given by

$$\mathcal{L}(\theta) = \frac{1}{N}\left(\sum_{x \in \mathcal{X}} \|\mathcal{N}_\theta(C'x) - x\|^2 + \lambda \|C'\mathcal{N}_\theta(C'x) - C'x\|^2\right). \tag{41}$$

For training, we used a regular `adam` optimizer. Due to the Transformer core we used a small learning rate of $3 \cdot 10^{-3}$ with a gentle decay of $0.97$ every $40,000$ iterations, and a batch size of 32. We trained the network for $800,000$ iterations using the same data used to train our models. We performed hyperparameter sweeps on the embedding dimension, the number of ResNet blocks, the number of self-attention layers, kernel sizes and regularization parameter $\lambda$. We observed that increasing $\lambda$ did not provide much overall performance boost, so the final version was trained with $\lambda$ equal to zero. Also, adding more blocks and self-attention layers saturated the performance quickly. For each combination of hyperparameters, the training took between 14 and 27 hours depending on the number of trainable parameters. The models used for the comparison in Table 2 have the following hyperparameters:

- $8\times$ **downscaling**: dimension embedding in the embedding layer - 16; number of downsampling blocks - 2; number of upsampling blocks - 5; number of ResNet block (in each of the up-sampling/downsampling) - 4; number of self-attention blocks - 2. Total number of parameters: $9,726,209$.
- $16\times$ **downscaling**: dimension embedding in the embedding layer - 32; number of downsampling blocks - 1; number of upsampling blocks - 5; number of ResNet block (in each of the up-sampling/downsampling) - 4; number of self-attention blocks - 2. Total number of parameters: $2,669,505$.

We point out that the network for the example with $8\times$ downscaling factor has more parameters due to the extra downsampling block which quadruples the number of embedding dimension. We also considered bigger dimension embedding, but we found no considerable gains in performance.

### D.4 cycleGAN

For the cycleGAN we followed closely the implementation in [86].

For the generators, we use the same architecture as in the original paper. The first layer performs a local embedding, followed by a sequence of downsample blocks, each of which downsamples

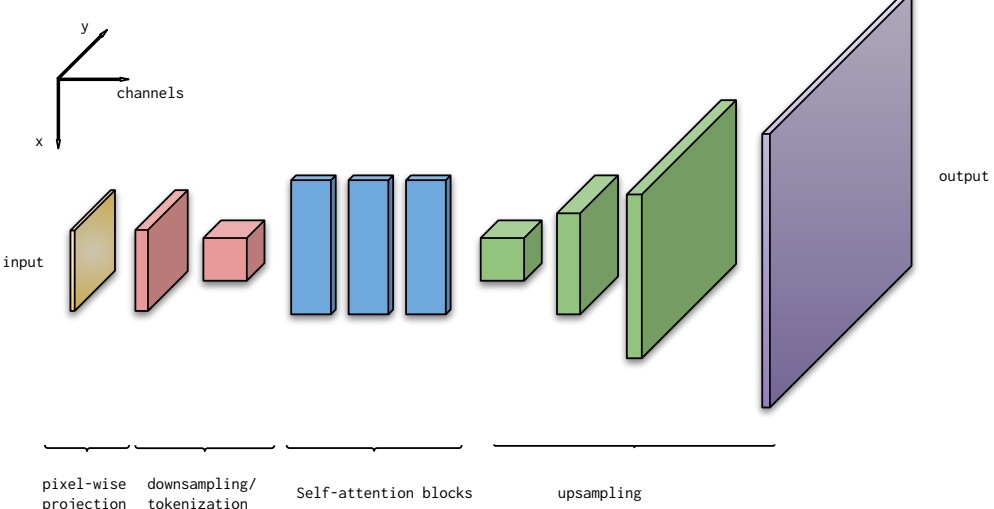

input        output

pixel-wise   downsampling/
projection    tokenization     Self-attention blocks      upsampling

Figure 5: Sketch of the structure of the deterministic upsampling network. In red we have the downsampling layers, which reduce the spatial resolution in space of the image, while increasing the number of channels. The first two channels are flattened into a one-dimensional set of tokens, and a sequence of self-attention blocks are applied afterwards. After the self-attention blocks, the tokens are reshaped back to it two-dimensional geometry, followed by a cascade of transpose convolutions to upsample the image.

the geometrical dimensions by a factor two, while increasing the channel dimension. At the lowest resolution we implement a sequence of ResNet blocks to process the input, immediately followed by a sequence of upsampling blocks, which upsample the geometrical dimension while reducing the channel dimension.

Given that the two generators, $\mathcal{G}_{\mathcal{X} \mapsto \mathcal{Y}}$ (from high-resolution to low-resolution) and $\mathcal{G}_{\mathcal{Y} \mapsto \mathcal{X}}$ (from low-resolution to high-resolution) have different input dimensions, we use a different combination of downsample/upsample blocks, and they also have different embedding dimensions. We implemented the different generators (for both the $8\times$ and $16\times$ downscaling factor) with different number of downsampling versus upsampling layers in the generators, and also different embedding dimensions.

Instead of using one discriminator architecture as in the original paper, we use two of them given that the input dimensions are different. Below we provide further details on the architecture used.

### D.4.1   Generator networks

**Embedding** We use one convolution layer with a kernel of size $(7, 7)$ and an embedding dimension that is different for each generator and for each problem.

**Downsampling blocks** We implement the downscaling blocks following [86]. These blocks effectively double the number of input channels while reducing the other dimensions by a factor of two. This downsampling is achieved using a convolutional layer with a $(2, 2)$ stride, a kernel of fixed width, and periodic boundary conditions. Subsequently, the output is normalized using a group normalization layer and further processed with a ReLU activation function.

**ResNet core** At the lowest resolution we use a sequence (whose length was also tuned) of ResNet blocks, using two convolution layers, with periodic boundary conditions, including a skip connection, two group normalization layers, and a dropout layer with a tunable dropout rate, following [86].

**Upsampling blocks** The upsampling blocks are implemented using transpose convolutional layers, followed by a group normalization layer and a ReLU activation function.

After several sweeps on the number of upsampling/downsampling blocks and other hyper-parameters we chose the following network configurations.

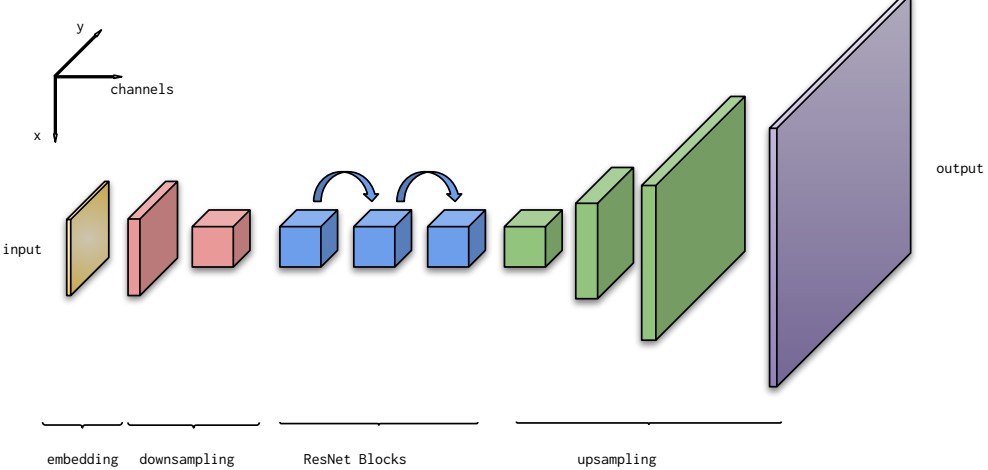

Figure 6: Sketch of the structure of the cycleGAN generator $\mathcal{G}_{\mathcal{Y}\mapsto\mathcal{X}}$. In red we have the downsampling layers, which reduce the spatial resolution in space of the image, while increasing the number of channels. At the lowest level we have (in blue) the ResNet blocks, followed by a cascade of transpose convolutions to upsample the image.

### $8\times$ downscaling factor

- $\mathcal{G}_{\mathcal{Y}\mapsto\mathcal{X}}$: number of downsampling blocks - 2; number of upsampling blocks - 5; embedding dimension - 32; dropout rate - 0.5; number of ResNet blocks - 6. Number of parameters: $4,029,569$.
- $\mathcal{G}_{\mathcal{X}\mapsto\mathcal{Y}}$: number of downsampling blocks - 5; number of upsampling blocks - 2; embedding dimension - 6; dropout rate - 0.5; number of ResNet blocks - 6. Number of parameters: $4,232,353$.

### $16\times$ downscaling factor

- $\mathcal{G}_{\mathcal{Y}\mapsto\mathcal{X}}$: number of downsampling blocks - 2; number of upsampling blocks - 6; embedding dimension - 64; dropout rate - 0.5; number of ResNet blocks - 6. Number of parameters: $16,868,641$,
- $\mathcal{G}_{\mathcal{X}\mapsto\mathcal{Y}}$: number of downsampling blocks - 5; number of upsampling blocks - 2; embedding dimension - 6; dropout rate - 0.5; number of ResNet blocks - 6. Number of parameters: $4,232,353$.

We point out that in this case the network for the $16\times$ example also requires more parameters: roughly four times more due to the higher dimension of the upsampling.

### D.4.2   Discriminator networks

The discriminator networks are the same as those in the original cycleGAN, with a small difference. The discriminator for $\mathcal{X}$ requires a special structure: instead of discriminating the full snapshot, we discriminate patches of the snapshot. By employing this trick, we were able to efficiently train the network, whereas using a global discriminator did not allow us to train the network to generate the snapshots as shown in §4.2. One simple strategy to implement this patched discriminator was to use the same architecture for both discriminators. However, we output a tensor of scores in which each element of the tensor corresponded to the score of one of the patches in the image, rather than a single score for the entire image. By choosing the patch size to be equal to the size of the lowest resolution snapshot, we could reuse the same architecture, depending on the problem size and downscaling factor.

The discriminator network, as described in [86], is composed of the following components: an embedding layer that applied a convolution with a kernel size of $(4, 4)$, a stride of two, padding of

one, and a tunable embedding dimension; a leaky ReLU applied with an initial negative slope of $0.2$; and a sequence of downsampling blocks similar to the generator network. Finally, a per-channel bottleneck network with one output channel is used to produce the local score.

The specific architectures used in Table 2 with their corresponding hyperparameters are summarized below:

- $8\times$ **downscaling factor.** The discriminators were the same: they had 3 downsampling blocks, with an embedding dimension of $64$. Total number of parameters $2,763,589$ each.
- $16\times$ **downscaling factor.** The discriminators were different due to the smaller dimensions of the snapshots in $\mathcal{Y}$, which would have resulted in very small receptive fields for the discriminator of $\mathcal{X}$. The discriminator for $\mathcal{Y}$ had two downsampling blocks, while the discriminator for $\mathcal{X}$ had six downsampling blocks. However, both discriminators had an embedding dimension of $64$. The total number of parameters was $15,349,576$ for the discriminator of $\mathcal{X}$ and $661,316$ for the discriminator of $\mathcal{Y}$.

### D.4.3 Loss and optimization

We also closely followed the original cycleGAN paper [86], in which we utilize the least-squares GAN loss in conjunction with the cycle loss. However, we do not employ the identity loss, as the different dimensions of the spaces make it challenging to impose such a loss naturally.

The optimization was performed by alternating the update of the generators and the discriminators. We used two `adam` optimizers: one for the generators and the other for the discriminators. Both optimizers had a momentum parameter $\beta$ set to 0.5 and a learning rate of 0.0002. Despite the continuous decrease in losses, we observed the emergence of several artifacts in the generated images. To address this issue, we checkpointed the model every two epochs, computed the MELR (see Equation Eq. (13)), and selected the model with the smallest unweighted error. For the example with an $8\times$ downscaling factor, this was achieved after just 8 epochs, while for the example with a $16\times$ downscaling factor, this was achieved after 16 epochs.

The full training loop took around two days to complete. However, due to early stopping, the checkpoints shown in Table 2 took around 8 hours to produce.

### D.5 ClimAlign

For this baseline we follow the original paper ClimAlign [32], in which the authors perform first an cubic interpolation and then use the AlignFlow framework to perform the debiasing.

For the implementation of the debiasing step we follow closely the implementation of the original AlignFlow [34] algorithm, which can be found in `https://github.com/ermongroup/alignflow`. We considered the same hyper parameters as in the original paper. The main modification we perform to the codebase was how to feed the data to the model.

## E Ablation studies

### E.1 Conditioning strengths

As described in section A, the parameter $\alpha$ controls the strength of conditioning in the subspace orthogonal to the linear constraint. Increasing its value encourages these orthogonal components to be more coherent with the constrained components, but at the same time makes sampling more costly. As such, we conduct grid searches to determine its value, along with the number of SDE solver steps, that strikes a satisfying balance between sample quality and cost.

The grid search setup is as follows: we first normalize $\alpha$ with respect to the dimensionality of $C'$

$$\tilde{\alpha} = \alpha/\gamma_{C'}, \quad \gamma_{C'} = \dim(\tilde{y})/\dim(x) \tag{42}$$

such that the same normalized $\tilde{\alpha}$ value does not have drastically different effects for different downscaling factors. Then we evaluate the unweighted MELR and sample variability for 2500 generated samples (50 conditions, 50 conditional samples each) resulting from combinations of $\tilde{\alpha} \in [0.125, 0.25, 0.375, ..., 3]$ and $N \in [32, 64, 128, ..., 1024]$.

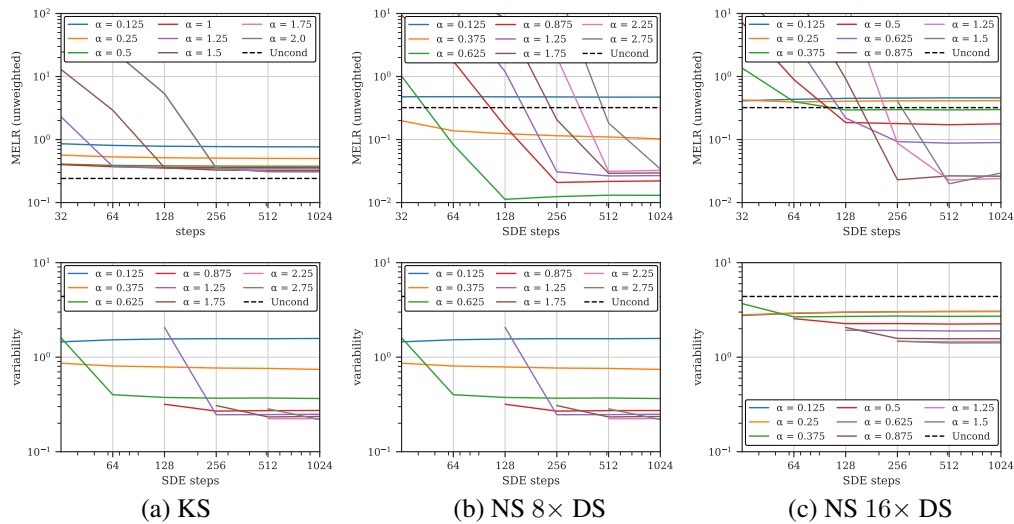

|         | (a) KS | (b) NS $8\times$ DS | (c) NS $16\times$ DS |
|---|---|---|---|

Figure 7: Unweighted MELR (Eq. (13); first row) and sample variability (Eq. (36); second row) vs. number of SDE steps at different values of $\tilde{\alpha}$. Larger $\tilde{\alpha}$ generally has better MELR but takes more solver steps to converge and results in lower sample variability.

Table 6: Best conditional sampling configurations and metrics found via grid search. For reference, the unconditional diffusion samples have variability 1.33 (KS) and 3.67 (NS); reference samples have variability 1.33 (KS) and 4.39 (NS); unconditional diffusion samples have unweighted MELR 0.27 (KS) and 0.37 (NS).

|  | KS | NS | | | |
|---|---|---|---|---|---|
|  | $8\times$ | $8\times$ | $16\times$ | $32\times$ | $64\times$ |
| % of conditioned elements ($\gamma_{C'}$) | 12.5 | 1.56 | 0.39 | 0.098 | 0.024 |
| Condition strength ($\tilde{\alpha}$) | 1.0 | 0.625 | 0.625 | 0.375 | 0.125 |
| SDE steps ($N$) | 256 | 256 | 512 | 1024 | 1024 |
| Sample variability | 0.04 | 0.36 | 1.56 | 3.52 | 3.67 |
| MELR (unweighted) | 0.36 | 0.06 | 0.05 | 0.06 | 0.21 |

In Fig. 7, we show MELR and variability plotted against $N$ for different $\tilde{\alpha}$'s. The MELR trends (first row) confirm our intuition that more steps are required for convergence as $\tilde{\alpha}$ increases. However, higher $\tilde{\alpha}$ also means lower sample variability (second row). This prompts us to choose an $\tilde{\alpha}$ that is neither too high nor too low. The selected configurations are listed in rows 2 and 3 of Table 6.

## E.2 Downscaling factors

We additionally obtain samples for $32\times$ and $64\times$ downscaling (conditioned values are obtained by further downsampling the OT corrected LR snapshots), besides the $8\times$ and $16\times$ presented in the main text, to explore the limits of our methodology. We conduct the same grid search as described in section E.1 to determine the normalized conditioning strength $\tilde{\alpha}$ and the number of solver steps $N$. The resulting configurations and metrics are displayed in Table 6, along with samples from an example test case in Fig. 8.

We observe that the variability of the generated conditional samples expectedly increases with the downscaling factor, as sampling process becomes less constrained. At $32\times$ downscaling, the corrected LR conditioning still plays a significant role in addressing the color shift in the unconditional sampler, leading to MELR resembling those obtained in the $8\times$ and $16\times$ cases. The same no longer holds true, however, for the $64\times$ downscaling case, as the MELR performance becomes more similar to that of the unconditional sampler. This outcome is also not surprising considering that $64\times$ downscaling corresponds to conditioning on $4 \times 4 = 16$ pixels, accounting for a minuscule 0.02% of the sample dimensions.

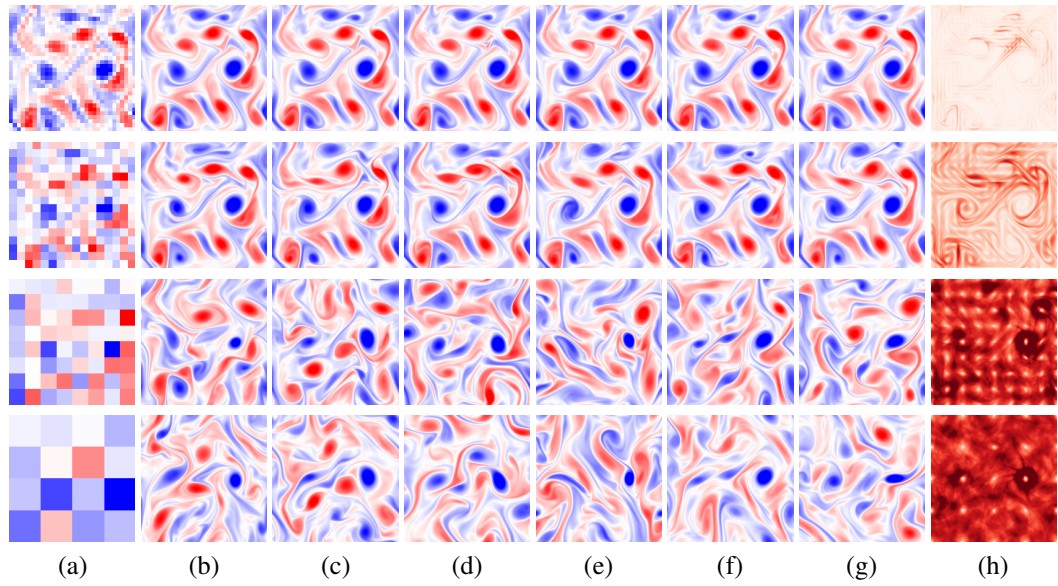

|     |     |     |     |     |     |     |     |
| (a) | (b) | (c) | (d) | (e) | (f) | (g) | (h) |

Figure 8: Sample comparison across different downscaling factors for NS ($8\times, 16\times, 32\times$ and $64\times$ for rows 1, 2, 3 and 4 respectively). Column legend: (a) conditioned values; (b-g) 6 samples generated by diffusion model conditioned on (a); (h) pixel-wise variability of 128 random conditional samples (same color scale across rows; dark means large and light means small).

Table 7: Metric comparison for conditioning on raw vs. OT-corrected LR snapshots for diffusion-, interpolation- and ViT-based super-resolution.

|  | Diffusion | | Cubic | | ViT | |
| --- | --- | --- | --- | --- | --- | --- |
|  | Raw | OT | Raw | OT | Raw | OT |
| MELR (unweighted), NS $8\times$ | 0.79 | **0.06** | 0.93 | **0.52** | 1.39 | **0.38** |
| MELR (unweighted), NS $16\times$ | 0.54 | **0.05** | 0.83 | **0.55** | 1.97 | **1.38** |
| MELR (weighted), NS $8\times$ | 0.37 | **0.02** | 0.41 | **0.06** | 0.58 | **0.18** |
| MELR (weighted), NS $16\times$ | 0.30 | **0.02** | 0.45 | **0.14** | 0.32 | **0.10** |

### E.3 Uncorrected super-resolution

To demonstrate the importance of debiasing the low-resolution data, we contrast the performance between conditioning on LR data before and after the OT correction in Table 7 for all factorized baselines considered. We observe that applying the correction universally leads to better samples regardless of the super-resolution method used.

## F Datasets

We consider two dynamical systems with chaotic behavior, which is the core property of atmospheric models [53]. In particular, we consider the one-dimensional Kuramoto-Sivashinsky (KS) equation and the Navier-Stokes (NS) equation with Kolmogorov forcing. For each equation we implement two different discretizations. The different discretizations are used to generate the low- and high-resolution data.

### F.1 Equations

**Kuramoto-Sivashinsky (KS) equation** We solve the equation given by

$$\partial_t u + u\partial_x u + \nu\partial_{xx}u - \nu\partial_{xxxx}u = 0 \qquad \text{in } [0, L] \times \mathbb{R}^+, \tag{43}$$

with periodic boundary conditions, and $L = 64$. Here the domain is rescaled in order to balance the diffusion and anti-diffusion components so the solutions are chaotic [28].

The initial conditions are given by

$$u_0(x) = \sum_{j=1}^{n_c} a_j \sin(\omega_j * x + \phi_j), \tag{44}$$

where $\omega_j$ is chosen randomly from $\{2\pi/L, 4\pi/L, 6\pi/L\}$, $a_j$ is sampled from a uniform distribution in $[-0.5, 0.5]$, and phase $\phi_j$ follows a uniform distribution in $[0, 2\pi]$. We use $n_c = 30$.

**Navider-Stokes (NS) equation** We also consider the Navier-Stokes equation with Kolmogorov forcing given by

$$\frac{\partial \boldsymbol{u}}{\partial t} = -\nabla \cdot (\boldsymbol{u} \otimes \boldsymbol{u}) + \nu \nabla^2 - \frac{1}{\rho}\nabla p + \mathbf{f} \qquad \text{in } \Omega, \tag{45}$$

$$\nabla \cdot \boldsymbol{u} = 0 \qquad \text{in } \Omega, \tag{46}$$

where $\Omega = [0, 2\pi]^2$, $\boldsymbol{u}(x, y) = (\boldsymbol{u}_x, \boldsymbol{u}_y)$ is the field, $\rho$ is the density, $p$ is the pressure, and $\mathbf{f}$ is the forcing term given by

$$\mathbf{f} = \begin{pmatrix} 0 \\ \sin(k_0 y) \end{pmatrix} + 0.1\boldsymbol{u}, \tag{47}$$

where $k_0 = 4$. The forcing only acts in the $y$ coordinate. Following [48], we add a small drag term to dissipate energy. An equivalent problems is given by its vorticity formulation

$$\partial_t \omega = -\boldsymbol{u} \cdot \nabla\omega + \nu\nabla^2\omega - \alpha\,\omega + f, \tag{48}$$

where $\omega := \partial_x \boldsymbol{u}_y - \partial_y \boldsymbol{u}_x$ [6], which we use for spectral method which avoids the need to separately enforce the incompressibility condition $\nabla \cdot \boldsymbol{v} = 0$. The initial conditions are the same as the ones proposed in [48].

### F.2 Pseudo-Spectral discretization

To circumvent issues stemming from dispersion errors, we choose a pseudo-spectral discretization, which is known to be dispersion free, due to the *exact* evaluation of the derivatives in Fourier space, while possessing excellent approximation guarantees [76]. Thus, few discretization points are needed to represent solutions that are smooth.

We used `jax-cfd` spectral elements tool box which leverages the Fast Fourier Transform (FFT) [19] to compute the Fourier transform in space of the field $u(x, t)$, denoted by $\hat{u}(t)$. Besides the approximation benefits of using this representation, the differentiation in the Fourier domain is a diagonal operator: it can be calculated by element-wise multiplication according to the identity $\partial_x \hat{u}_k = ik\hat{u}_k$, where $k$ is the wavenumber. This makes applying and inverting linear differential operators trivial since they are simply element-wise operations [76].

The nonlinear terms in Eq. (43) and Eq. (48) are computed using Plancherel's theorem to pivot between real and Fourier space to evaluate these terms in quasilinear time. This procedure transforms Eq. (43) and Eq. (45) to a system in Fourier domain of the form

$$\partial_t \hat{u}(t) = \mathbf{D}\hat{u}(t) + \mathbf{N}(\hat{u}(t)), \tag{49}$$

where $\mathbf{D}$ denotes the linear differential operators in the Fourier domain and is often a diagonal matrix whose entries only depend on the wavenumber $k$ and $\mathbf{N}$ denotes the nonlinear part. We used a 4th order implicit-explicit Crack-Nicolson Runge-Kutta scheme [11], where we treat the linear part implicitly and the nonlinear one explicitly.

### F.3 Finite-volumes discretization

We use a simple discretization using finite volumes [51], which was implemented using the finite volume tool-box in `jac-cfd` [48]. For the KS equation, we used a Van-Leer scheme to advect the field in time. This was implemented by applying a total variation diminishing (TVD) limiter to the Lax-Wendroff scheme [51]. The Laplacian and bi-Laplacian in Eq. (43) were implemented using tri-

and penta-diagonal matrices. The linear systems induced by the implicit step were solved on-the-fly at each iteration using fast-diagonalization.

For the NS equation we used a fractional method, which performs an explicit step which relies in the same Van-Leer scheme for advecting the field together with the diffusion step. We then performed a pressure correction by solving a Poisson equation, also using fast-diagonalization by leveraging the tensor structure of the discretized Laplacian.

### F.4 Data Generation

For the low-fidelity, low-resolution data (specifically the space $\mathcal{Y}$ in Fig. 1), we employed the finite-volume schemes described above, using either a fractional discretization in time (for NS) or a implicit-explicit method (for the KS equation). The domains mentioned above were utilized, with a $32 \times 32$ grid and a time step of $dt = 0.001$ for NS. For KS, we employed a discretization of size $48$ points and a time step of $dt = 0.02$. These resolutions represent the lowest settings that still produced discernible trajectories.

For the high-fidelity, high-resolution data (namely the space $\mathcal{X}$ in Fig. 1), we used the pseudo-spectral discretization mentioned above with a $256 \times 256$ grid and time step $dt = 0.001$ for NS (using the vorticity formulation) and discretization of size $192$ and time step $dt = 0.0025$ for the KS equation.

For the KS equation, we created $512$ trajectories in total. Each trajectory was run for $4025$ units of time, of which we dropped the ones generated during an initial ramp-up time of $25$ units of time. Of the remaining $4000$ units of time, we sampled each trajectory every $12.5$ units of time resulting on $320$ snapshots per trajectory.

For NS we also created $512$ trajectories in total. We used the same time discretization for both low- and high-resolution data. Each trajectory was run for $1640$ units of time, of which we dropped the ones generated during an initial ramp-up time of $40$ units of time. Of the trajectories spanning the remaining $1600$ units of time, we sampled each them every $4$ units of time (or $4000$ time steps) resulting on $400$ snapshots per trajectory.

The sampling rate for each trajectory was chosen to minimize the correlation between consecutive snapshots and therefore, obtain a better coverage of the attractor.

## G Hyperparameters

Table 8 shows the set of hyperparameters used to train our diffusion models. Our U-Net model (parameterizing $F_\theta$ in Eq. (31)) closely follows the *Efficient U-Net* architecture in [67] and apply self-attention operations at the coarsest resolution only. We employ the standard `adam` optimizer, whose learning rate follows a schedule consisting of a linear ramp-up phase of 1K steps and a cosine decay phase of 990K steps. The maximum learning rate is $10^{-3}$ and the terminal learning rate is $10^{-6}$. We additionally enable gradient clipping (i.e., forcing $\|dL/d\theta\|_2 \leq 1$) during optimization.

## H Debiasing with optimal transport

We begin this section by giving an overview of computational methods to find optimal transport maps.

For certain measures, the optimal transport plan $\gamma \in \Pi(\mu_Y, \mu_{Y'})$ in the Wasserstein-2 distance $W_2(\mu_Y, \mu_{Y'}) = \inf_\gamma \int \frac{1}{2}\|y - y'\|^2 d\gamma(y, y')$ is induced by a transport map $T: \mathcal{Y} \mapsto \mathcal{Y}'$ where $T_\sharp \mu_Y = \mu_{Y'}$. In particular for the quadratic cost, Brenier's theorem guarantees that such a map exists when $\mu_{Y'}$ is atom-less [8] and the plan is concentrated on the graph of a map, i.e., $\gamma(y, y') = (\mathrm{Id}, T)_\sharp \mu_{Y'}$. Moreover, the Brenier map $T$ is given by the gradient of a convex potential function.

Recently, several methods have been proposed to approximate the Brenier map given only a collection of i.i.d. samples from each measure $\{y^i\} \sim \mu_Y, \{(y')^i\} \sim \mu_{Y'}$. These include flow-based models [77], the projection arising from an entropic-regularized OT problem as discussed in Section [65], and continuous approximations of discrete plans [63]. Another recent approach directly parameterizes the transport map as the gradient of a convex potential function that is represented using input convex neural networks [50, 55]. This approach leverages the dual formulation of the OT problem, to express

Table 8: Hyperparameters for diffusion model architecture and training.

| Hyperparameter | KS | NS |
|---|---|---|
| Input dimensions | $192 \times 1$ | $256 \times 256 \times 1$ |
| `Dblock`/`Ublock` resolutions | $(96, 48, 24)$ | $(128, 64, 32, 16)$ |
| Resolution channels | $(32, 64, 128)$ | $(32, 64, 128, 256)$ |
| Number of `ResNetBlocks` per resolution | 6 | 6 |
| Noise embedding | Fourier | Fourier |
| Noise embedding dimension | 128 | 128 |
| Number of attention heads | 8 | 8 |
| Total number of parameters | 4.40M | 31.44M |
| Batch size | 512 | 16 |
| Number of training steps | 1M | 1M |
| EMA decay | 0.95 | 0.99 |
| Training duration (approximate) | 2 days | 4 days |

the Wasserstein-2 distance as

$$W_2(p, q)^2 = C_{p,q} + \sup_{f \in \mathrm{cvx}(p)} \left\{ \mathbb{E}_p[-f(X)] + \mathbb{E}_q[-f^*(Y)] \right\}, \tag{50}$$

where $C_{p,q} = \mathbb{E}[X^2] + \mathbb{E}[Y^2]$ is a constant and $f^*(y) = \inf_x \{x^T y - f(x)\}$ is the convex conjugate of $f$. Under the conditions of Brenier's theorem, the optimal map $T$ satisfying $T_\sharp \mu_Y = \mu_{Y'}$ corresponds to $T = \nabla f^*$ where $f$ solves Eq. (50). If we replace $f^*$ with a second network $g$ that is also parameterized with ICNNs, [55] proposed to find the OT map by solving the min-max problem:

$$\sup_{f \in \mathrm{cvx}(p)} \inf_{g \in \mathrm{cvx}(q)} \left\{ \mathbb{E}_p[-f(X)] + \mathbb{E}_q[-\langle Y, \nabla g(Y) \rangle - f(\nabla g(Y))] \right\}.$$

This approach is very sensitive to the network initialization and is challenging to solve in high-dimensions due to the constraints imposed on the map. Moreover, they are limited to squared-Euclidean costs, which limits their flexibility in certain applications. As a result, in our numerical examples we choose to use the entropic OT problem discussed in Section 3.3.

### H.1 Additional numerical results

We provide additional numerical results to showcase how the optimal transport (OT) map corrects the bias in the LFLR snapshots.

Fig. 9 displays how the OT map changes the covariance structure of the snapshots, while Fig. 10 shows the cumulative distribution functions before and after the OT correction for both the $8\times$ and $16\times$ NS downscaling problems. We can observe from the plots that the OT successfully corrects the distributions.

## I   Computational resources

The generation of the data was performed using 12 core server with an NVIDIA A100 GPU and 40 GB of VRAM. The training for the diffusion models, and the ViT model were performed in a 16 core server with NVIDIA V100 GPUs with 32 GB of VRAM. The cycle-GAN was trained on a TPU v4 in Google cloud. The Sinkhorn iteration for computing the OT map was performed in a 80 core instance with 240GB of RAM, each training loop took roughly a day for 5000 iterations. All the training was performed in single precision (`fp32`), while the generation of the data was performed in double precision (`fp64`). The data was transferred to single precision at training/inference time.

## J   Additional samples

We provide additional conditional samples in Figs. 11 and 12 from the NS $8\times$ and $16\times$ downscaling experiments respectively.

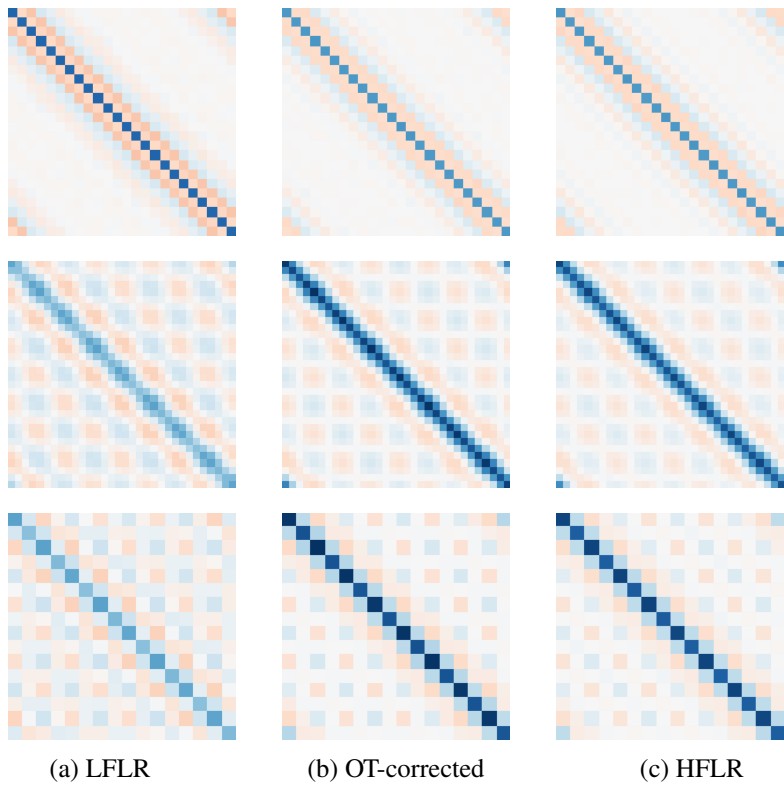

(a) LFLR       (b) OT-corrected       (c) HFLR

Figure 9: Covariance structure of LFLR, OT-corrected and HFLR reference samples for KS (top) NS $8\times$ downscaling (middle) and NS $16\times$ downscaling (bottom).

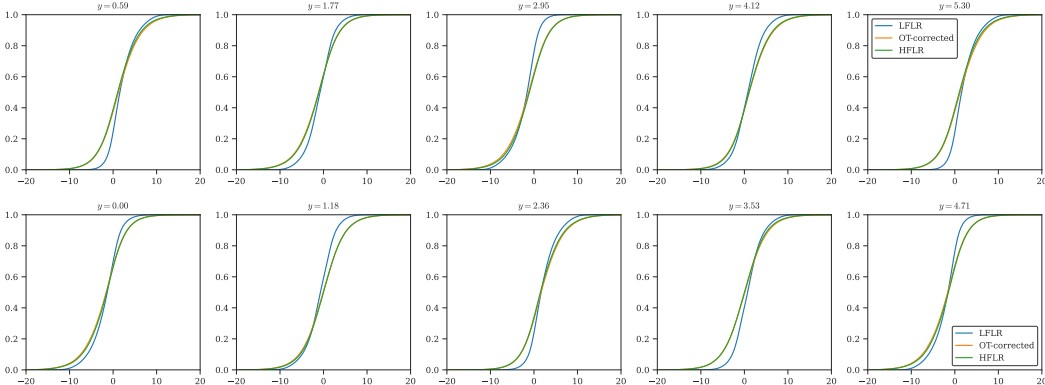

Figure 10: Cumulative distribution functions (CDFs) at selected locations of the snapshots for the NS $8\times$ (top) and $16\times$ (bottom) examples.

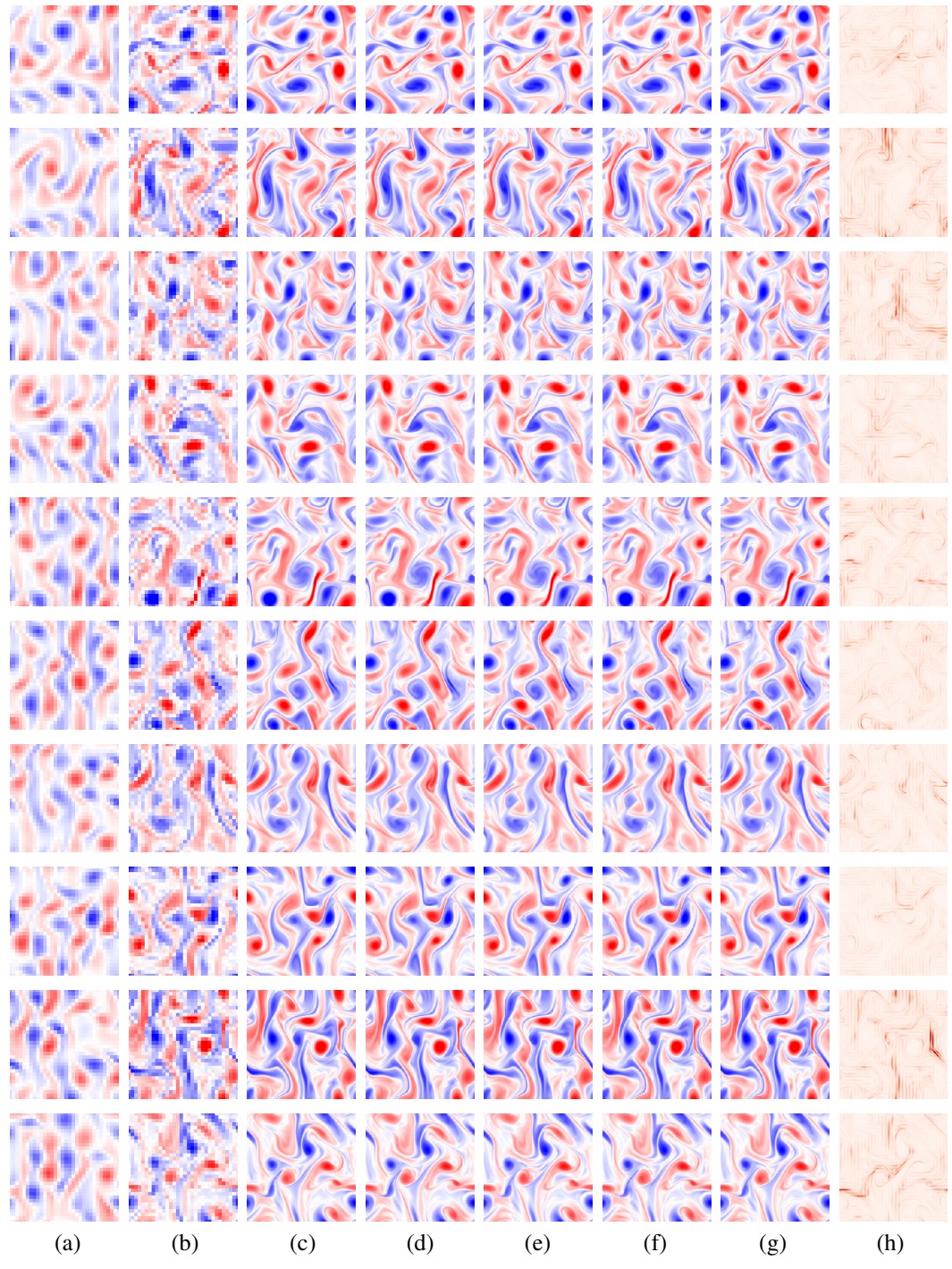

(a)     (b)     (c)     (d)     (e)     (f)     (g)     (h)

Figure 11: Conditional samples for NS $8\times$ downscaling. Column legend: (a) raw LFLR snapshot; (b) LFLR snapshot corrected by OT; (c-g) 5 samples generated by diffusion model conditioned on (b); (h) pixel-wise variability of 128 random samples conditioned on (b).

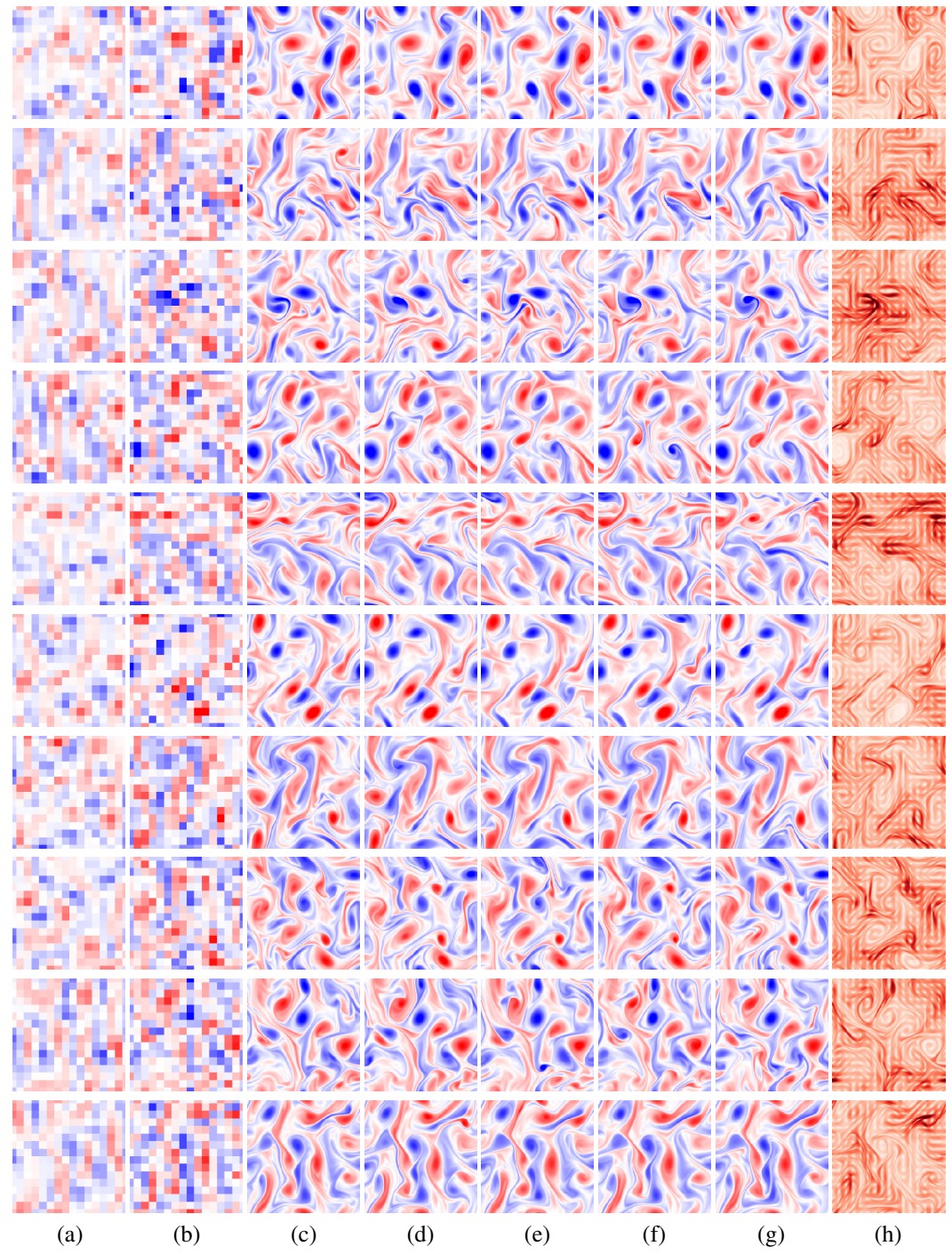

(a)       (b)       (c)       (d)       (e)       (f)       (g)       (h)

Figure 12: Conditional samples for NS $16\times$ downscaling. Column legend: (a) raw LFLR snapshot; (b) LFLR snapshot corrected by OT; (c-g) 5 samples generated by diffusion model conditioned on (b); (h) pixel-wise variability of 128 random samples conditioned on (b).