# OpenReview forum: "Debias Coarsely, Sample Conditionally: Statistical Downscaling through Optimal Transport and Probabilistic Diffusion Models"
_NeurIPS.cc/2023/Conference — NeurIPS 2023 spotlight_

### Official Review · Reviewer_zUt7 · 2023-06-28

**Soundness:** 2 fair
**Presentation:** 4 excellent
**Contribution:** 2 fair
**Rating:** 5
**Confidence:** 3

**Summary:**

This paper proposed a new two-stage method for statistical downscaling by combining a coarse de-biasing step based on optimal transport and a conditional up-sampling step based on a diffusion model.

**Strengths:**

Overall, the paper is very well written and clearly explains the proposed method. The proposed method is divided into two components in a straightforward manner and is justified clearly. Empirical results confirm the applicability of the proposed method and its superior performance compared to several baselines such as cycle-GAN and ViT based super-resolution.

**Weaknesses:**

The main novelty of this work is the introduction of a debiasing step for downscaling applications due to the biased nature of the problem. But Figure 3 shows the debiased result can be quite different from LR data. Because the diffusion model is very dependent on the debiased result, this debiasing step is crucial for the success of accurate downscaling in my opinion. Comparatively, cycle-GAN seems to output a more similar result to the LR data. So it seems necessary to discuss further if this debiasing step is indeed correcting the bias in LR data or introducing additional error/dissimilarity.

**Questions:**

1. Using diffusion models for super-resolution has been widely studied, so it would be helpful to clarify if Section 3.2 is related to any relevant works such as SNIPS and DDRM? If so then please cite them in the paper.
2. What is Constraint RMSE in Table 2? And why is cycle GAN not included in this comparison? From my above comment, such a comparison could be helpful.

**Limitations:**

The authors have addressed limitations of the work adequately. Broader impacts are not applicable for this work.

---

> ### Author Rebuttal · Authors · 2023-08-09
>
> Thank you for your positive review. Please find our response below.
>
> > Because the diffusion model is very dependent on the debiased result, this debiasing step is crucial for the success of accurate downscaling in my opinion.
>
> Thank you for your sharp observation that the debiased results from OT are less close to the low resolution inputs than the output of cycle-GAN. We believe this is a feature, not a bug. While appearing counterintuitive, there are several reasons that this type of **“visual/pixel-based similarity” might not be preferred**.
>
> First, the goal of debiasing is to transform the low-resolution data to a high-resolution one such that the latter lies close to the manifold of the high-resolution data (so that diffusion models can be sampled through posterior constraints), rather than just super-resolving/upsampling the low-resolution sample.  In our response to Review wBNS, we used the analogy to restore a distorted image due to an aberrative lens or distorting mirror. There, the desirable outcome would be an image that is close to the original (high-res, undistorted) image but less so to the distorted version. Note that, since our data is unpaired, there is no point-wise measurement of similarity to a single high-res image - instead, we have to resort to measuring distribution differences.  Our results do show that while visually the debiased images look different from the LR ones, **in distribution, they are close to the high resolution ones**.
>
> Secondly, cycle-GAN’s results might indicate a weakness of the method in solving the problem considered in this paper. The debiased outputs stay close to the low-resolution images so that they do not explore enough to match the samples from the high-res manifold.  We do not think this is the design goal of cycle-GAN; the method’s assumption does not take into consideration that the bias can be large enough so that the low-resolution and high-resolution manifolds can be apart significantly. Also, we point out that as the downscaling factor become higher, and therefore the downscaling becomes more delocalized and therefore, it contains a bigger bias, cycle-GAN struggles to produce realistic images as seen in Fig. 3 c) in the manuscript.
>
> To have a more thorough comparison, we added two new baselines and metrics to the suite of experiments, as shown in the Table found in the general response. We use one of the new baselines, Bias Correction and Spatial Downscaling (BCSD), to showcase the need for non-local debiasing that is able to nudge the samples from one manifold to another. BCSD, is a popular technique in statistical downscaling. In a nutshell, it performs a cubic interpolation, and then a debiasing step based on pixel-wise quantile matching. The latter ensures the correct pixel-wise statistics of the downscaled output, as shown by the very low pixel-wise Wasserstein-1 error in the Table. We can observe in Figure 1c) of the included PDF that the resulting downscaled image preserves most of the geometrical information of the low-resolution input. However, all the spatially-dependent metrics are much worse for this method. This indicates that this marginal de-biasing method fails to "nudge" the low-resolution input towards the correct high-resolution manifold.
>
> > Using diffusion models for super-resolution has been widely studied, so it would be helpful to clarify if Section 3.2 is related to any relevant works such as SNIPS and DDRM? If so then please cite them in the paper.
>
> Thank you for the references. We were not aware of them. Our approach is indeed related, although less intrusive in the unconditional score function. Also, as shown in [1] our approach can be also used for non-linear constraints, even though such property was not used in the current manuscript. We will add the references to the manuscript.
>
> > What is Constraint RMSE in Table 2? And why is cycle GAN not included in this comparison? From my above comment, such a comparison could be helpful.
>
> The constraint was not added because cycleGAN does not hinge on conditional sampling with respect to the low-resolution debiased snapshot ($y’$ in Fig. 1) using a user defined downsampling map. We will add a comparison with respect to the original low-fidelity low-resolution snapshot.
>
> References:
>
> [1]: User-defined Event Sampling and Uncertainty Quantification in Diffusion Models for Physical Dynamical Systems
> M. A. Finzi, A. Boral, A. G. Wilson, F. Sha, and L. Zepeda-Núñez, International Conference on Machine Learning, 10136-10152

---

> > ### Comment · Reviewer_zUt7 · 2023-08-14
> >
> > Thank the authors for their detailed response. Like Reviewer wBNS, my main concern is still the optimal transport step of the proposed method. While I understand in theory the points raised in the response ('feature, not a bug'; 'visual/pixel-based similarity' might not be preferred), it seems that these arguments are weak and not supported by concrete evidence. Indeed, while I agree the optimal transport map does transport between the biased and debiased manifolds, it is not clear to me if the learned map is indeed close to 'optimal'. It appears that the added metrics do not reflect this aspect. However, please feel free to reply and correct any points I made.

---

> > > ### Author Response · Authors · 2023-08-15
> > >
> > > Thank you for your response.
> > >
> > > First, we would like to clarify the nomenclature “optimal” in this case.  The map is optimal in the sense that it satisfies optimality conditions of a minimization problem averaged upon the full distribution, subject to the hard constraints that the marginals are respected. In particular, preserving these constraints is fundamental to correcting statistical biases, but it makes the problem much harder. Furthermore, the map presented here is an approximation via an entropic regularization, which renders the optimization problem tractable. In our formulation, OT allows us to find maps directly between the Y and Y’ spaces, when the biased and unbiased distributions are only prescribed using samples. On the other hand, other transport maps (which are not based on optimality conditions) often need to specify an intermediate distribution (e.g., a standard Gaussian latent space), which results in complex maps that need to be approximated when the distributions on Y and Y’ are highly non Gaussian even if they are similar.
> > >
> > > Second, we want to point out that Table 1 in our main text shows that OT effectively corrects statistical biases in the low-resolution data with respect to various metrics. As additional evidence to support the OT-based debiasing map outperforms the others, we have downsampled the BCSD, cycGAN and ClimAlign baselines to the low resolution space so they can be directly compared to the output of the OT component in our method. We have computed additional distribution metrics (covariance error, uMELR, wMELR, MMD) for these baselines and attached the resulting metrics in the table below. It is clear that OT yields superior performance in all of these metrics. As mentioned in our first response, this comes at the “cost” of pixel-wise similarity to some degree, which we quantified through the pointwise symmetric mean absolute percentage error (sMAPE) metric in the attached table (measures how much the explicit/implicit debiasing moves the low resolution samples on average by computing the relative $\ell^1$ distance between the input low-resolution sample $y$ and its debiased output for each method). We observe that OT does not move significantly more than other baselines (except for BCSD, which in fact solves the OT problem independently at each pixel, but does not respect pixel correlations). We hope this addresses your concern sufficiently well.
> > >
> > > |              | OT   | BCSD | cycGAN | ClimAlign |
> > > |--------------|------|------|--------|-----------|
> > > | 8xdownscale  |      |      |        |           |
> > > | cov          | 0.08 | 0.31 | 0.16   | 2.21      |
> > > | uMELR        | 0.01 | 0.95 | 0.08   | 0.53      |
> > > | wMELR        | 0.03 | 0.13 | 0.04   | 0.54      |
> > > | MMD          | 0.04 | 0.06 | 0.06   | 0.61      |
> > > | sMAPE        | 0.53 | 0.25 | 0.41   | 0.74      |
> > > | 16xdownscale |      |      |        |           |
> > > | cov          | 0.08 | 0.35 | 0.33   | 2.50      |
> > > | uMELR        | 0.02 | 0.63 | 0.34   | 0.67      |
> > > | wMELR        | 0.03 | 0.16 | 0.15   | 0.58      |
> > > | MMD          | 0.03 | 0.34 | 0.09   | 0.55      |
> > > | sMAPE        | 0.54 | 0.36 | 0.63   | 0.76      |

---

### Official Review · Reviewer_Z19Y · 2023-07-06

**Soundness:** 4 excellent
**Presentation:** 4 excellent
**Contribution:** 4 excellent
**Rating:** 8
**Confidence:** 5

**Summary:**

The authors proposed a two-stage probabilistic framework for unpaired data. The problem is factorized into two steps, an optimal transport (OT) based mapping for debiasing and a diffusion-based model for up-sampling. The problem is demonstrated on fluid mechanics datasets representing difficult fluid and weather problems. The predicted results matched the statistics of the physical properties well.

**Strengths:**

The current paper developed a statistical downscaling framework for unpaired data. Tackling this problem is very crucial in learning from multi-scale, multi-fidelity models, especially for large-scale applications like weather/climate modeling.

Originality:  The idea of correcting the low-frequency bias by an OT map is novel and interesting. Moreover, the OT map is also integrated into the SOTA diffusion model framework. Due to the gradient information in diffusion modeling, a posterior conditioning sampling can further improve the performance at inference time and satisfy the given constraints well. The two-stage factorization is novel since it doesn't require a cycle-consistency type of loss and allows computing the debiasing map in a lower dimensional space.

Quality: The numerical results show the developed model has a superior performance compared to the baseline models. The developed model also has the ability to provide reasonable uncertainty estimation. Moreover, comprehensive ablation studies and training details are provided to help evaluate the model.

Clarity: The paper is well-written and clearly guides the reviewer to understand it. Math is accurate, and adequate proof and derivation are provided.


Significance: The current work tackles the statistical downscaling problem in the weather/climate model. The ability to improve the accuracy of high-resolution forecasts from low-fidelity data has a broader impact in practical applications, like real-time weather forecasting.

**Weaknesses:**

Some additional details and explanations are needed. See the question part.

**Questions:**

1. What is the Reynolds number of the NS equations.

2. Are there any difficulties applying it to real-world turbulence data set?

3. In Figure 2 (a), what does the true trajectory look like?
(2) What does the corrected in Figure 2(b) mean? Does it mean OT correction only? Moreover, it is hard to distinguish the OT+cDfn, corrected, true, and UncondDfn. Therefore, it is hard to see the advantage of OT+cDfn from the figure.

4. What is the training cost?

**Limitations:**

The computational cost of OT mapping can be further reduced, which leads to a future research direction.

---

> ### Author Rebuttal · Authors · 2023-08-09
>
> We greatly appreciate your positive feedback. Please find our response below.
>
> > What is the Reynolds number of the NS equations?
>
> The Reynolds number is 1000. The (high-fidelity) simulation setup is identical to [1].
>
> > Are there any difficulties applying it to real-world turbulence dataset?
>
> Conceptually, we don’t expect any difficulty. However, in practice, we envision that the biggest hurdle would be the amount of data necessary to obtain an accurate debiasing step, unless some symmetries are exploited. In our example we used the Kolmogorov flow, which is known to be ergodic with an unknown but relatively low-dimensional manifold, which we can cover with a relatively small amount of samples. For a truly turbulent flow this condition may not hold, and therefore, a large amount of data may be required.
>
> > In Figure 2 (a), what does the true trajectory look like? (2) What does the corrected in Figure 2(b) mean? Does it mean OT correction only? Moreover, it is hard to distinguish the OT+cDfn, corrected, true, and UncondDfn. Therefore, it is hard to see the advantage of OT+cDfn from the figure.
>
> We would like to point out that only snapshots are considered, not trajectories. Assuming that the former is what is asked, we want to note that even at the dimensionality of the KS system, it is prohibitive to draw “ground truth” conditional samples (e.g. via rejection sampling) that can be used to compare against the ones shown in Figure 2(a).
>
> “Corrected” does mean OT correction only. The figure is meant to provide a sense of the qualitative nature of the methods. It tends to emphasize the method’s ability to capture large-scale features. To further distinguish them, the energy and covariance metrics are especially informative as they give quantitative measures for the small-scale features not easily visible from the figure.
>
> > What is the training cost?
>
> Training the unconditional network required about a day in a V100 GPU. The tuning of the sampling parameters took roughly one day. The debiasing, which is the most time consuming part of the algorithm due to the large memory footprint that makes GPU acceleration hard using an off-the-shelf method, took roughly three days. Using GPU acceleration and on-the-fly matrix-vector products we believe it should be possible to reduce the training time of the debiasing map significantly. One could also take advantage of several advances in computational optimal transport seeking to reduce the memory footprint of this computation. These include low rank and sparse approximations to the optimal transport plan/map, as in [2, 3,4].
>
> References:
>
> [1] Kochkov, Dmitrii, et al. "Machine learning–accelerated computational fluid dynamics." Proceedings of the National Academy of Sciences 118.21 (2021): e2101784118.
>
> [2] Low-rank Optimal Transport: Approximation, Statistics and Debiasing, Meyer Scetbon, Marco Cuturi, NeurIPS 2022.
>
> [3] Approximating Optimal Transport via Low-rank and Sparse Factorization, Weijie Liu, Chao Zhang, Nenggan Zheng, Hui Qian, 2021.
>
> [4] Monge, Bregman and Occam: Interpretable Optimal Transport in High-Dimensions with Feature-Sparse Maps, Marco Cuturi, Michal Klein, Pierre Ablin, ICML 2023.

---

> > ### Comment · Reviewer_Z19Y · 2023-08-19
> > **Reviewer response**
> >
> > Thanks for the author’s detailed response and additional baselines and metrics results. After reading other reviews, I find the concerns are addressed satisfactorily by the elaboration and the empirical evaluation. The motivation is elaborated, and I was convinced that dealing with unpaired data is practical and challenging in climate systems. And the OT is a good approach to achieve debiasing and map data between different manifolds. The OT+diffusion can improve the sample quality compared with diffusion only. The strong empirical evidence showed that the proposed method outperforms all the baselines for versatile metrics. I believe it is a solid work and can bring insights into the ML+climate science community. Overall, I’d be happy to recommend a strong acceptance.

---

### Official Review · Reviewer_wBNS · 2023-07-07

**Soundness:** 2 fair
**Presentation:** 2 fair
**Contribution:** 2 fair
**Rating:** 3
**Confidence:** 3

**Summary:**

The authors suggest a simple approach for the problem of statistical downsampling, which is the super-resolution of low-resolution weather grids. The approach involves first "debias-ing" the low-resolution grid via solving an optimal transport problem, then obtaining a high resolution image by solving an image super-resolution problem with a score-based diffusion model.

**Strengths:**

**Novel application of diffusion models.** The authors appear to apply diffusion modeling to a novel problem, even though the application appears to be straightforward.

**Promising results.** According to the metrics provided by the authors, the results are promising. (However, I am concerned about the validity of the metrics.)

**Weaknesses:**

**Unclear motivations.** Since this is predominantly a climate modeling paper submitted to a machine learning conference, the posed problem is certainly unfamiliar to me, and probably unfamiliar to most readers. The authors need to clearly describe the motivation of the problem. How can statistical downscaling ever succeed in reproducing the high fidelity, high resolution outputs if the model is so chaotic that the initial conditions do not even matter (Lines 32-34, and Footnote 1, Page 6)?

**Validity of "debiasing".** The authors propose to reduce the discrepancy between low-resolution and high-resolution weather grids via optimal transport. This assumes some kind of continuity of the grids, i.e. similar weather systems in the lower dimensional weather grids produce similar simulated trajectories, that also correspond to similar weather systems in the higher dimensional weather grids. But a central assumption that necessitates the practice of using "unpaired" data is the "discontinuity" of the problem. Similar weather systems do not produce similar trajectories (Lines 32-34, and Footnote 1, Page 6). Therefore, is debiasing even possible? Moreover, is optimal transport the correct approach?

**Concerns with empirical evaluations.** This paper suggests that the proposed method obtains better performance according to their metrics, which appear to measure the deviation of the modeled statistics with the ground truth statistics. This comparison seems somewhat unfair, since the proposed super-resolution model is trained solely by modeling the statistics of the ground truth data. No other method except for the CycleGAN implementation even attempts to similarly model the ground truth distribution $\mathcal{X}$, and even CycleGAN has auxiliary (i.e., cycle consistency) losses. This is corroborated by Table 1: only CycleGAN approaches the performance of the proposed method. No competing deep learning-based approaches to statistical downsampling are used in the comparison. Moreover, the metrics used for evaluation differ greatly from [1], which appears to be a well-cited paper in deep learning-based statistical downsampling.

**Clarity in writing and formatting.** There are multiple places where the writing quality affects the readability of the text.

5: "tandeming" Tandem is not a verb and feels awkward here, consider using a different word?

14-16: "Moreover, our procedure correctly matches the statistics of physical quantities, even when the low-frequency content of the inputs and outputs do not match, a crucial but difficult-to-satisfy assumption needed by current state-of-the-art alternatives." What does this sentence mean?

21-22: "Consequentially, accurate predictions ... need to be *downscaled* from *coarser lower-resolution* models' outputs." (*Emphasis* mine.) Since this paper is submitted to a machine learning conference, where topics in computer vision are much more common than those in climate modeling, the authors need to clearly delineate where *downscaling* corresponds to statistical downscaling in the climate modeling sense, versus downscaling in the computer vision sense, especially when their meanings are completely reversed. I spent a lot of time trying to parse this sentence.

Figure 1 and its caption is unclear. Y' is mentioned here but it is not explained until Section 3.

Citation links do not work.

[1] DeepSD: Generating High Resolution Climate Change Projections through Single Image Super-Resolution. https://arxiv.org/pdf/1703.03126.pdf

**Questions:**

See "Weaknesses" section.

**Limitations:**

See "Weaknesses" section.

---

> ### Author Rebuttal · Authors · 2023-08-09
>
> Thank you for your detailed review, especially your comments on places we could have explained better.
>
> **Problem Setting, Motivation and Validity**
>
> The problem we study in this paper is analogous to image or video super-resolution on a high level, but it has several important distinctions. We appreciate the opportunity to clarify this point.
>
> Weather and climate are examples of an inherently chaotic dynamical system with an (approximately) stationary distributions. All snapshots of the system state may be considered samples drawn from the system’s stationary distribution. The proverbial [Lorenz butterfly](https://en.wikipedia.org/wiki/Lorenz_system) is such an example of the (low-dimensional) attractor of a (simplified) dynamical system used to describe the weather and climate. While two trajectories can be close at one time, they could be quite far from each other at later times while still belonging to the same stationary distribution over the attractor. In this aspect, the initial condition does not matter as the stationary distribution “forgets” the initial condition. Thus, the objective of statistical downscaling is to recover the stationary distribution, embodied by its samples.
>
> The bias is introduced because different numerical schemes (e.g., with different integration order, step sizes, etc.) yield different perturbed versions of the stationary distribution on a (possibly different) attractor. When this happens, samples from two different distributions do not have a correspondence and hence can only be seen as unpaired samples from different attractors.
>
> The central problem we hope to resolve is: given one sample X from a stationary distribution over an attractor A, can we obtain a set of representative samples from the stationary distribution over the attractor B, where A and B stems from the underlying dynamical system?
>
> You are correct that it is impossible to find the sample that corresponds to X, as there is no such correspondence to begin with (unless we assume infinite precision). Instead, we identify the conditional distribution of samples from B that correspond to X – in other words, if we sample X from the distribution over A, and we collect all the samples from the conditional distribution, then we can recover the samples of the distribution over B that correspond to X. In this sense, debiasing is possible.
>
> OT is the approach we consider to debias. A key contribution of our approach is to recognize the need to debias and propose a two-stage factorization for downscaling. Since we need to compare distributions (so as to debias), directly incorporating the debiasing map into the diffusion model is challenging as one would need to sample first and compute the distribution discrepancy measure (say, a kernel maximum mean discrepancy) and then differentiate through the score network to learn the debiasing map. We attempted such an approach, but found the computational cost to be prohibitive. Using OT we decouple the two steps by learning the debiasing map without the need to sample from the diffusion model.
>
> If we could use image super-resolution as an analogy, the bias could be seen as a distortion to the original high-resolution sample (say using aberrative lens or distorting mirror) while downsampling to a lower-resolution. Thus, the super-resolution would need to unwarp the distortion in the low-resolution first and then upsample.
>
> **Metrics and Baselines**
>
> Thanks for the suggestions. Please see our general response and response to reviewer wBNS.
>
> **Other Questions**
>
> > 5: "tandeming" Tandem is not a verb and feels awkward here, consider using a different word?
>
> Acknowledged. We will modify this sentence.
>
> > 14-16: "Moreover, our procedure correctly matches the statistics ..." What does this sentence mean?
>
> We agree that this sentence is hard to read. We wanted to contrast our methodology to other approaches such as [1], in which the method needs the low-frequency component of the input and output to match (i.e., the large scale features in both low and high resolution distributions match). Thus most of the debiasing is performed in the medium- to high-frequency regime (i.e., only the medium and small features), which renders the problem much easier. Our methodology does not enforce this constraint. The lack of this constraint makes the problem more challenging to solve, but it broadens the applicability of the methodology. We will clarify the context of this sentence, and add the reference.
>
> > 21-22: "Consequently, accurate predictions ... " ... I spent a lot of time trying to parse this sentence.
>
> To clarify, we use downsampling as the opposite to upsampling (or super-resolution), following the typical ML-jargon. For downscaling, however, we refer not only upsampling/super-resolution but also debiasing - this is a standard term used in weather/climate community.
>
> We hope that the explanation above has made this difference clear. This is in contrast with super-resolution in which only upsampling is considered, which we explain in Fig. 1. A typical upsampling (or super-resolution) method will seek to extrapolate in frequency the red line, which would provide results that are not in the correct distribution. Thus, we are required to debias the input (go from the red to the blue line) and then upsample it. This process is what we refer to as downscaling.
>
> > Figure 1 and its caption is unclear. Y' is mentioned here but it is not explained until Section 3.
>
> Thank you for the observation. We will add a definition in the caption for $y’ \in \mathcal{Y}’$.
>
> > Citation links do not work.
>
> We will make sure this is fixed.
>
> References:
>
> [1] DeepSD: Generating High Resolution Climate Change Projections through Single Image Super-Resolution. https://arxiv.org/pdf/1703.03126.pdf

---

> > ### Comment · Reviewer_wBNS · 2023-08-20
> >
> > Thank you very much for your response. I appreciate work the authors put into elucidating their points, and addressing my comments. However, I still lack some clarity on the two of the main concerns I initially raised in the review.
> >
> > 1) **Validity of debiasing**: In the rebuttal, the authors justify debiasing via a discussion on the stationary systems of weather and climate systems. I have trouble understanding this framework involving stationary distributions, as it seems ill-fit for describing weather and climate systems. Stationary systems are time-invariant. Aren't weather systems inherently variant over time (which is why predicting them at future times are of interest)? At any rate, optimal transport appears to me to still pose a strong assumption on the structure of the weather systems, which have highly "discontinuous" dynamics with respect to their initial state: similar weather systems do not produce similar trajectories (Lines 32-34, and Footnote 1, Page 6). Theoretically, why should optimal transport provide the correct "debiasing" solution?
> >
> > 2) **Metrics**: I could not find any discussion on most of the concerns I raised, namely this part:
> > > This comparison seems somewhat unfair, since the proposed super-resolution model is trained solely by modeling the statistics of the ground truth data. No other method except for the CycleGAN implementation even attempts to similarly model the ground truth distribution
> > , and even CycleGAN has auxiliary (i.e., cycle consistency) losses. This is corroborated by Table 1: only CycleGAN approaches the performance of the proposed method. No competing deep learning-based approaches to statistical downsampling are used in the comparison.
> >
> > Additionally, the authors mention that further discussion is provided here:
> > > Please see our general response and response to reviewer wBNS.
> > I am reviewer wBNS. I assume the authors mean reviewer zUt7?
> >
> > I still could not find the discussion on the above statement.
> >
> > For these reasons, I am still hesitant to raise my score.

---

> > > ### Author Response · Authors · 2023-08-20
> > > **first part of the comment**
> > >
> > > Thank you for reiterating your concerns and giving us the opportunity to clarify further. Please find our point-by-point response below.
> > >
> > > > Aren't weather systems inherently variant over time (which is why predicting them at future times are of interest)?
> > >
> > > The systems considered in the current work are regarded as being close to an [ergodic dynamical system](https://en.wikipedia.org/wiki/Ergodicity) which admits a stationary distribution that can be sampled by evolving the system in time (a close analogy would be some Markov chains having a invariant distribution). While over a few steps the system is time-variant as its state changes, the set of all states visited by the system over a long horizon is not.
> > >
> > > Perhaps the simplest example to showcase these properties (as mentioned in our previous response) is the Lorentz system, which is a  simplified mathematical model for atmospheric convection. The system is non-stationary in time (it relies on solving a time-dependent ODE), but if you sample snapshots (without the time-stamp) of a trajectory (or set of trajectories) for a long enough time, the samples will follow a given invariant distribution, which is often referred to as the Lorenz butterfly.
> > >
> > > Climate systems (not weather, which is intrinsically transient) fall into this category approximately. One needs to sample from really long time spans (at least 10s of years in real time) to have a workable coverage of the underlying distribution.
> > >
> > > In our work, we are concerned with two such stationary distributions - one ground truth with samples from the true system evolutions and the other consisting of samples from an approximation model (i.e. low-order PDE solver) with errors.
> > >
> > > We are not sure if we were able to answer your question. If you feel that our response was not clear or if we missed the point of your question, would you be able to rephrase your question? We would be glad to answer it as soon as we can.
> > >
> > > > At any rate, optimal transport appears to me to still pose a strong assumption on the structure of the weather systems, which have highly "discontinuous" dynamics with respect to their initial state: similar weather systems do not produce similar trajectories (Lines 32-34, and Footnote 1, Page 6).
> > >
> > > The "discontinuous"/chaotic property mentioned highlights the difficulty of the “paired data” setup - even if there is correspondence in the initial conditions (from the two distributions considered): this correspondence will eventually get lost over time. This is why in our view learning a sample-to-sample mapping **paired via time** is not feasible. We adopt the “unpaired” data setup and instead attempt to learn the many-sample-to-many-sample (i.e. distributional) mapping. As stressed above, a critical assumption is that the system admits a stationary distribution and is sufficiently sampled. We are exploiting this particular property to re-establish the “pairedness” in the data via OT.
> > >
> > > > Theoretically, why should optimal transport provide the correct "debiasing" solution?
> > >
> > > The solution to the distribution mapping problem is not unique. OT treats the matching of the distributions as a constraint and additionally imposes a “cost” associated with the map and explicitly minimizes it. This “cost” is a measure of the deviation of $T(x)$, the map applied to $x$, from its input $x$.  Intuitively, this means that we want the transport map to move the states **as little as possible** - in other words, the mapped state should still “look like” the original in some sense (here by minimizing the $L^2$ norm). We choose this cost to re-establish the paired relationship in the data. It is by no means the only constraint one can impose (e.g. cycGAN consistency enforces cycle consistency between the transports, while ClimAlign imposes the invertibility of the transport), but we believe it is the most intuitively sensible way.
> > >
> > > In order to showcase this issue we have added a table in the [response](https://openreview.net/forum?id=5NxJuc0T1P&noteId=vufkqwLpfq) to reviewer zUt7. The table provides a comparison of the deviation and matching of the distributions for the distribution mapping method benchmarked, under as many applicable metrics as we can think of. Compared to the other baselines considered, OT is able to best match the marginals, i.e., mapping one distribution to another, while having a relatively small deviation from the input. We would be happy to add any other applicable metric that you would like us to include, to the final manuscript.

---

> > > > ### Author Response · Authors · 2023-08-20
> > > > **second part of the comment**
> > > >
> > > > > No other method except for the CycleGAN implementation even attempts to similarly model the ground truth distribution
> > > >
> > > > We would like to point out that besides CycleGAN the two additional baselines that we added during rebuttal are both of this nature. BCSD [1] attempts to match the ground truth distribution independently at the pixel level, and it is the most popular method of debiasing in the weather and climate community, as mentioned by reviewer fyEE. ClimAlign [2] is based on constructing a normalizing flow between the two distributions directly.
> > > >
> > > > > No competing deep learning-based approaches to statistical downsampling are used in the comparison.
> > > >
> > > > As mentioned above, among all the benchmarked methods, we have two deep-learning based approaches learning the distributions directly, namely CycleGAN and ClimAlign (whose performance seriously degrades as we increase the downscaling-factor). Thus, we believe this is a fair comparison – please feel free to suggest any other applicable methods and we would be happy to add them into the revised version.
> > > >
> > > > Please note that we are *not* arguing which deep learning methods are the best. We are proposing that instead of simply upsampling the low-resolution inputs using a super-resolution method, debiasing in a separate stage is a useful approach.
> > > >
> > > > > Please see our general response and response to reviewer wBNS. I am reviewer wBNS. I assume the authors mean reviewer zUt7?
> > > >
> > > > We apologize for the confusion. Please ignore this reference and we hope our response above addresses your concerns.
> > > >
> > > > **References: **
> > > >
> > > > [1] Andrew W. Wood, Edwin P. Maurer, Arun Kumar, and Dennis P. Lettenmaier.  Long-range experimental hydrologic forecasting for the eastern United States, Climate and Dynamics 2002.
> > > >
> > > > [2] Brian Groenke, Luke Madaus, and Claire Monteleoni. ClimAlign: Unsupervised statistical downscaling of climate variables via normalizing flows, CI2020: Proceedings of the 10th International Conference on Climate Informatics, 2020.

---

> > > > > ### Author Response · Authors · 2023-08-20
> > > > > **extra response**
> > > > >
> > > > > Given that we are not sure if we answered the question appropriately, we would like to comment a bit further.
> > > > >
> > > > > > Aren't weather systems inherently variant over time (which is why predicting them at future times are of interest)?
> > > > >
> > > > > In particular, we would like to clarify again that our work do not **forecast**. Instead we debias and up-sample (or super-resolve) a stationary distribution, which is captured in a relatively short period of time (in contrast to centuries where external forcing such as CO2 could change underlying stationary distribution).
> > > > >
> > > > > In their comment, we believe the reviewer may be asking about "predictability" where weather/climate eventually drifts, characterized by a slow moving climatological distribution (often modeled as a stationary distribution that **slowly** drifts), versus whether we can recover high-resolution snapshots of states, from lower resolution weather and climate simulation.  As mentioned above, we study the latter problem. In this vein, we argue the system admits an approximate (in our rebuttal) stationary distribution, namely, the trajectories, low- or high-resolutions, are sampled from a given distribution (not a distribution moving non-stationarily). In particular, we are not seeking to use low-res climate data at time period $i$, to forecast high-res climate data at future time period $j$. Instead, we seek to use low-res climate data at time period $i$ to create high-res climate data at the same time-period $i$.
> > > > >
> > > > > We point out that the main plausible reason for the slow non-stationarity of climate models come from time-dependent external forcing (such as CO2 change due to anthropogenic activities) that slowly shifts the distribution. But in our work, we do not consider this case, instead we focus on relatively shorter times.
> > > > >
> > > > > > At any rate, optimal transport appears to me to still pose a strong assumption on the structure of the weather systems, which have highly "discontinuous" dynamics with respect to their initial state: similar weather systems do not produce similar trajectories (Lines 32-34, and Footnote 1, Page 6).
> > > > >
> > > > > "discontinuous dynamics":  we think the reviewer might be thinking of climate going through a sudden change --- but that is not what we study.  For a given time period $i$, we do assume the trajectories form a predictable distribution -- that is the basis of ensemble forecasts in weather and climate community in the last few decades.  Those trajectories cannot be precisely aligned (unless the time period is very small) due to the chaotic behavior of the dynamics. But the distance between two trajectories (and they corresponding snapshots) themselves is not an especially relevant quantity (except being two samples from a distribution).
> > > > >
> > > > > We hope that we have answered your concerns.

---

### Official Review · Reviewer_fyEE · 2023-07-26

**Soundness:** 4 excellent
**Presentation:** 4 excellent
**Contribution:** 4 excellent
**Rating:** 8
**Confidence:** 3

**Summary:**

This work introduces a new framework to tackle statistical downscaling, a climate science equivalent of super-resolution, in two steps: The first step removes the bias while staying at low resolution with an optimal transport method and the second step increases the spatial resolution with a diffusion-based model. The performance of the method is shown on two fluid-dynamics datasets: 2d Navier-Stokes and 1d Kuramoto-Sivanshinski equation. The proposed method outperforms baselines such as CycleGAN and ViT on all suggested metrics.

**Strengths:**

Originality: This work develops a new method to tackle statistical downscaling

Quality: Well written and well setup experiments.

Clarity: Good motivation and explanation for the two-step approach tackling this problem.

Significance: This work addresses an important and impactful problem and has high practical societal relevance. Especially, large upsampling factors like used here make the problem hard to solve and motivate the need for advanced methods like the one proposed here. The work addresses a probabilistic formulation of statistical downscaling, which is still under-researched.


**Weaknesses:**

This is a great paper, but some weaknesses are:
- The metrics used are not common in statistical downscaling. Very common metric that could be included is e.g. Continuous Rank Probability Score, see e.g. https://agupubs.onlinelibrary.wiley.com/doi/10.1029/2022MS003120 for more commonly used climate downscaling metrics
- No comparison with existing work like ClimAlign (https://arxiv.org/pdf/2008.04679.pdf), BCSD
- References to some relevant works are missing: https://arxiv.org/pdf/2211.16116.pdf (stat. downscaling using diffusion models)
- The work is motivated by climate/weather modeling but doesn’t include real climate/weather datasets. There are datasets available that include to different simulations at lower and higher resolution. It would be great to see how this method performs on real-world data. Datasets that could be used are NorESM data at different resolution or WRF simulations. Or learning the mapping from ERA-interim (https://climatedataguide.ucar.edu/climate-data/era-interim) to WRF (https://rda.ucar.edu/datasets/ds612.0/#!) data, like done in the ClimAlign work.


**Questions:**

- It would be great to motivate the selection of metrics more and explain why other, more common metric like CRPS are not used here
- A discussion at the beginning of the paper talking about different kinds of statistical downscaling setups such as perfect prognogsis (see e.g. https://www.cambridge.org/core/books/statistical-downscaling-and-bias-correction-for-climate-research/4ED479BAA8309C7ECBE6136236E3960F)
- To be applied in climate modeling it is also relevant to know the inference runtime (not only training time).

References:
I would suggest including more of existing climate super-res/statistical downscaling work. A extensive collection of DL for statistical downscaling literature can be found here: https://github.com/paulaharder/deep-downscaling-overview

Minor:
Why are don't the first two rows in Table 2 have the best scores in bold?

Typos:
Figure 1 caption: "an invertible" instead of "a invertible"
Line 52: "an unknown" instead of "a unknown"
Line 129: "a structured" instead of "an structured"
Line 141: "an unconditional" instead of "a unconditional"
Figure 2 caption: "and and" typo
Line 309: "s" missing of "sample"

**Limitations:**

I think the limitations of this work could be made clearer. Either in an additional limitations section or in the conclusion should be a mention that the method has only been applied to an idealized datasets and not a real climate model dataset. Also, it should be mentioned that this work tackles a specific subfield of statistical downscaling and other subfields, eg. perfect prognosis where predictors differ

---

> ### Author Rebuttal · Authors · 2023-08-09
>
> We greatly appreciate your detailed review. Please see below the responses to the issues raised.
>
> > Use common metrics in statistical downscaling such as CRPS, Motivate the use of the metrics listed in the paper
>
> Thank you for the comment. We will provide a more thorough explanation for the evaluation metrics considered in this work. Our main consideration was to assess the effectiveness of debiasing by measuring differences in statistics between the generated and true samples.
>
> CRPS is designed to assess an ensemble forecast with respect to a ground-truth deterministic observation, which does not quite fit our setup with unpaired data. Nonetheless, we have considered a somewhat clumsy application of the CRPS metric: treating each true sample as the deterministic observation and all generated samples as the ensemble forecast; the reverse is also performed treating each generated sample as the observation and all true samples as the ensemble. The resulting metrics are averaged and shown in the general response above. The results indicate that this symmetrized CRPS metric has little discrimination power across all elements. This is unsurprising considering that CPRS may be decomposed into a bias and a spread component and the latter completely dominates the metric - even comparing the ground truth distribution with itself leads to a large reference value for the metric.
>
> That said, we are adding two new metrics, namely the maximum mean discrepancy (MMD) between the real and generated high-resolution distributions and the pixel-wise Wasserstein-1 metric. These are common distribution metrics and we hope that they will help to provide a more comprehensive evaluation of our proposed method.
>
> > Comparison to existing work like ClimAlign, BCSD
>
> We thank the reviewer for the suggestion. We had tried to run ClimAlign using the code provided in the original GitHub repository, but were unable to adapt it to our setup directly due to code version and numerical issues. In particular, we found that the code was prone to produce singular matrices inside the GlowFlow module. We were not able to solve the problem and instead reimplemented the original AlignFlow code following [this Github repository](https://github.com/ermongroup/alignflow). Metrics are included in the general response.
>
> BCSD was implemented and added as a baseline (see general response). The preliminary results suggest that it is not competitive in general due to the lack of spatial correlations beyond the cubic interpolation result. However, the table above shows that the Wasserstein-1 metric is very small. This is expected because the algorithm uses a pixel-wise quantile matching procedure, which is the solution to the $L^1$ OT problem and gives rise to the Wasserstein-1 metric.
>
> Finally, we would like to comment that our methodology can be thought of as a generalization of BCSD using modern ML-tools.
> Our method considers covariant structures through the solution of  a global OT problem using entropy-regularized Sinkhorn iterations,  instead of solving an OT-problem marginally to each pixel by matching pixel-wise quantiles. Our method leverages diffusion models to model high-dimensional joint distribution that maintain spatial coherence, instead of using polynomial interpolation. We point out that,  even though this connection can be made for the resulting algorithm, our formulation hinges on a two-step factorization of the probabilistic description of statistical downscaling recasted as a sampling problem, which is not present in BCSD.
>
> > Missing References, including more existing work,
>
> Thank you for the pointer to the deep-downscaling-overview repo - it is a wonderfully curated source. We will add selected references.
>
> Regarding https://arxiv.org/pdf/2211.16116.pdf: we will add it to the bibliography. The authors tackle the problem in the setting of paired data. Thus, this is more analogous to super-resolution.
>
> Regarding perfect prognosis: Perfect prognosis is out of the scope of this paper (e.g., dynamical downscaling was also out of scope) as it still requires paired data between different predictors to learn empirical relationships. We will add a few sentences in the introduction providing some background and rendering the scope of this contribution more transparent viz-à-viz other statistical downscaling setups.
>
> > Inference runtime
>
> The inference runtime is around 1700 samples/min (in batches of 128) for KS and 20 samples/min (in batches of 16) for NS. We will add this information to the text.
>
> > I would suggest including more of existing climate super-res/statistical downscaling work.
>
> Thank you for the references! We will go over the list and add relevant ones to the final version of the paper.
>
> > Clearer discussion on the limitations of this work
>
> We agree and will remind readers that the current paper mainly demonstrates the plausibility of the proposed methodology (i.e., the factorization of the downscaling process) for a particular instance of the statistical downscaling problem. Subfields of statistical downscaling are certainly very important and we will discuss those in the final version of the paper. In particular, we will discuss how to extend the algorithmic framework to tackle those problems.
>
> While turbulence data provides a strong proof of concept, we are actively working on real climate/weather datasets. Preliminary results suggest the algorithmic framework is robust. We look forward to hearing your suggestions and comments.

---

### Author Rebuttal · Authors · 2023-08-09

**General Rebuttal Response**

We thank all reviewers for providing such detailed reviews. We are encouraged by the comments that the current idea has novelty, presented in a clear way, and that the results have potentially high (societal) impact and relevance.

To address the weaknesses and questions raised in the comments, we want to highlight the main changes below:
* **Added baselines.** We have added two more baselines - *Bias Correction and Spatial Downscaling (BCSD)* and *ClimAlign*. The former is a popular method for statistical downscaling consisting of an upsampling step using cubic interpolation followed by a pixel-wise bias correction using quantile matching. The latter is a neural network approach based on normalizing flows coupled with a cyclic regularization step.
* **Added metrics.** We have also added two more metrics: the *mean minimum discrepancy (MMD)* and a mean pixel-wise *Wasserstein-1* distance. We show that our methodology outperforms, or at least remains competitive, in these new metrics compared to the old and new baselines.
* **Writing.** We thank all reviewers again for catching typos and phrasing issues. Most of these are acknowledged in our response to individual reviewers. We will make sure to fix them in the next version.
* **Figure and Table updates.** The updated metric table (replacing Table 2) is attached below. In the attached PDF, we show the updated sample comparison (Figure 3) and energy plots (Figure 2c and 4c) that include the newly added baselines.

Furthermore, we would like to re-emphasize the fact that our methodology **targets unpaired data**. Consequently, we do not have access to the ground truth conditional (i.e., posterior) samples, which are computationally prohibitive to obtain using methods like rejection sampling. This is the primary reason that the metrics we used may differ from metrics in similar works (e.g. RMSE, CRPS) that assume access to paired data during training and/or evaluation. To this end, we focus on metrics based on distributional differences, as a way to measure the fidelity of the generated samples. This is analogous to the way in which the fidelity of class- or text-conditioned image generation methods are evaluated using unconditional metrics like FID.

**Updated metric table**
| Model             | Var  | covRMSE↓ | MELRu↓ | MELRw↓ | KLD↓  | Wass1↓ | MMD↓ |
|-------------------|------|----------|--------|--------|-------|--------|------|
| **8x downscale**  |      |          |        |        |       |        |      |
| BCSD              |   0  |   0.31   |  0.67  |  0.25  |  2.19 |  0.23  | 0.10 |
| cycGAN            |   0  |   0.15   |  0.08  |  0.05  |  1.62 |  0.32  | 0.08 |
| ClimAlign         |   0  |   2.19   |  0.64  |  0.45  | 64.37 |  2.77  | 0.53 |
| Raw+cDfn          | 0.27 |   0.46   |  0.79  |  0.37  | 73.16 |  1.04  | 0.42 |
| OT+Cubic          |   0  |   0.12   |  0.52  |  0.06  |  1.46 |  0.42  | 0.10 |
| OT+ViT            |   0  |   0.43   |  0.38  |  0.18  |  1.72 |  1.11  | 0.31 |
| (ours) OT+cDfn    | 0.36 |   0.12   |  0.06  |  0.02  |  1.40 |  0.26  | 0.07 |
| **16x downscale** |      |          |        |        |       |        |      |
| BCSD              |   0  |   0.34   |  0.67  |  0.25  |  2.17 |  0.21  | 0.11 |
| cycGAN            |   0  |   0.32   |  1.14  |  0.28  |  2.05 |  0.48  | 0.13 |
| ClimAlign         |   0  |   2.53   |  0.81  |  0.50  | 77.51 |  3.15  | 0.55 |
| Raw+cDfn          | 1.07 |   0.46   |  0.54  |  0.30  | 93.87 |  0.99  | 0.39 |
| OT+Cubic          |   0  |   0.25   |  0.55  |  0.13  |  7.30 |  0.85  | 0.20 |
| OT+ViT            |   0  |   0.14   |  1.38  |  0.09  |  1.67 |  0.32  | 0.07 |
| (ours) OT+cDfn    | 1.56 |   0.12   |  0.05  |  0.02  |  0.83 |  0.29  | 0.07 |

**Results for symmetric CRPS**

(To show that CRPS does not properly discriminate - see response to reviewer fyEE for context)

|      | Reference | BCSD | cycGAN | Raw+cDfn | OT+Cubic | OT+ViT | OT+cDfn |
|------|-----------|------|--------|----------|----------|--------|---------|
| CRPS |    2.50   | 2.25 |  2.42  |   1.87   |   2.47   |  2.18  |   2.53  |

---

> ### Author Response · Authors · 2023-08-18
> **Thank you and rebuttal feedback.**
>
> We would like to thank again the reviewers for their time and reviews. They have been helpful to improve and refine the manuscript.
> Given that the discussion period is finishing soon, we are eager to hear any further comments, including feedback on the rebuttal,

---

### Decision · Program_Chairs · 2023-09-21

**Decision:**

Accept (spotlight)

**Comment:**

This paper considers the problem of statistical downscaling, a common task in climate informatics, by sequentially debiasing (based on optimal transport) and upsampling (based on diffusion models). The reviewers agree in general that the approach is innovative, well-motivated, and effectively tested, and provides an effective solution to an important problem. Questions were raised about the motivation for the approach; however, I believe these have been adequately addressed by the authors. Overall, I believe the paper merits a strong acceptance.